# Expressivity-Efficiency Tradeoffs for Hybrid Sequence Models

John Cooper [* 1]  Ilias Diakonikolas [1]  Mingchen Ma [* 1]  Frederic Sala [1]

## Abstract

Hybrid sequence models—combining Transformer and state-space model layers—seek to gain the expressive versatility of attention as well as the computational efficiency of state-space model layers. Despite burgeoning interest in hybrid models, we lack a basic understanding of the settings where—and underlying mechanisms through which—they offer benefits over their constituent models. In this paper, we study this question, focusing on a broad family of core synthetic tasks. For this family of tasks, we prove the existence of fundamental limitations for non-hybrid models. Specifically, any Transformer or state-space model that solves the underlying task requires either a large number of parameters or a large working memory. On the other hand, for two prototypical tasks within this family—namely selective copying and associative recall—we construct hybrid models of small size and working memory that provably solve these tasks, thus achieving the best of both worlds. Our experimental evaluation empirically validates our theoretical findings. Importantly, going beyond the settings in our theoretical analysis, we empirically show that learned—rather than constructed—hybrids outperform non-hybrid models with up to $6\times$ as many parameters. We additionally demonstrate that hybrid models exhibit stronger length generalization and out-of-distribution robustness than non-hybrids. [1]

## 1 Introduction

Transformers are the workhorse architecture for modern language models. While highly expressive and capable, Transformer-based models suffer from high complexity, par-

ticularly for inference time processing of long sequence inputs. As a result, developing alternative non-Transformer architectures has become among the most important problems in LLM development. Structured state space models (SSMs) like Mamba (Gu & Dao, 2024) are among the most promising such alternatives. Such models trade off complexity for expressivity (Jelassi et al., 2024), achieving higher throughput—but typically lower performance—compared to Transformer-based models.

A natural question is whether we can sidestep this trade-off and produce a model architecture that offers the best of both worlds. *Hybrid sequence models* seek to achieve this objective. These models, which mix layers from Transformer architectures (e.g., attention layers) with SSM layers, ideally outperform either Transformer-only or SSM-only models. In a short time, hybrids that can empirically do so on particular tasks have been scaled up from tiny sizes to as large as 50 billion parameters. For example, Nvidia's Nemotron-H hybrid model family (Blakeman et al., 2025) offers both better downstream evaluation performance *and* higher throughput (due to the presence of lower-complexity Mamba layers) than Transformer-only baselines.

Despite these empirical successes, ***we have no principled understanding*** of why hybrid models can outperform models made up of a single type of layer. Similarly, we do not yet know for what basic tasks we should expect hybrids to behave in this way. This paper takes the first steps towards providing a ***fundamental theory addressing architectural tradeoffs for hybrid models***. It does so by first showing that on a family of core tasks where *pure* (i.e., standard Transformer-only and SSM-only) models provably suffer from limitations (in terms complexity and memory). In contrast, we build constructions of hybrid models that *do not* have the same limitations on representative tasks—including key tasks like associative recall and selective copying—thus exhibiting provable benefits for hybrids.

Concretely, we evaluate the performance of a model by analyzing its *input-independent memory (model size) and input-dependent memory (working memory)*. We focus on a *function-composition* family of tasks (Fig. 1) that combine both a long-context control variable and a local context-addressable lookup; such tasks naturally model real-world data. For these, (i) under an injectivity condition, we prove

---

[1]Department of Computer Science, University of Wisconsin, Madison. Correspondence to: John Cooper <jfcooper2@cs.wisc.edu>, Mingchen Ma <mingchen@cs.wisc.edu>.

*Proceedings of the $43^{rd}$ International Conference on Machine Learning*, Seoul, South Korea. PMLR 306, 2026. Copyright 2026 by the author(s).

[1]Code is available at this link

that, for this family, any pure SSM requires large internal state (or many layers); as a result, their size scales linearly with respect to the hidden dimension of the problem to solve the problem. Likewise, (ii), under a global-sensitivity condition, any sliding-window Transformer (which includes full-window attention) requires a large window scaling linearly with respect to the length of the input context. Together, this pair of results indicates that for a broad class of tasks, pure SSM-based models and pure Transformer-based models *fail to achieve good expressivity and inference efficiency simultaneously*. We study two representative synthetic tasks in this family, namely selective copying and a variant of associative recall (Arora et al., 2023). For these tasks, we construct ***provably successful shallow hybrid models*** whose size scales with the logarithm of the size of the tasks while using only sublinear memory.

Empirically, we validate our theoretical results and investigate hybrid versus non-hybrid performance in further settings and on additional tasks, such multi-key associative recall (MKAR) and needle-in-a-haystack (NH). We find that for selective copying and MKAR, ***hybrids can perform the task with similar or better quality than the pure models with 6 times fewer parameters***. For associative recall with decoding, at the tested scales, the pure models *never* match the performance possible with the hybrid model. On top of measuring performance at fixed model sizes, we observe that hybrid models exhibit stronger length generalization and out-of-distribution (OOD) robustness. We see that when trained on the same distribution of short examples, hybrids consistently out-perform pure Transformers by around 10% accuracy for long sequences. For out-of-distribution testing, the hybrid model sometimes attains over 15% higher performance than either the Transformer or the SSM with around the same number of parameters.

**Roadmap of the paper.** In Section 2, we provide necessary preliminaries and notations. In Section 3, we introduce a family of tasks formulated as computing a function composition and provide conditions under which non-hybrid models fail to solve the tasks efficiently. Next, in Section 4, we focus on two specific tasks (of varying difficulty) that are within this family and construct hybrid models that outperform non-hybrids. In Section 5, we conduct experiments to show the benefits of hybrid models empirically.

## 2 Preliminaries and Notations

We provide necessary preliminaries and notation. A complete list of preliminaries is deferred to Section B.

We consider sequence-to-sequence token prediction problems. Let $\mathcal{V}$ be some vocabulary of tokens, $V = |\mathcal{V}|$, and $\vec{\mathbf{x}} = (\mathbf{x}_i)_{i=1}^L$ be an input sequence. A language model $M$ a is sequence-to-sequence map $M : \mathcal{V}^L \to \mathcal{V}^m$ of the form

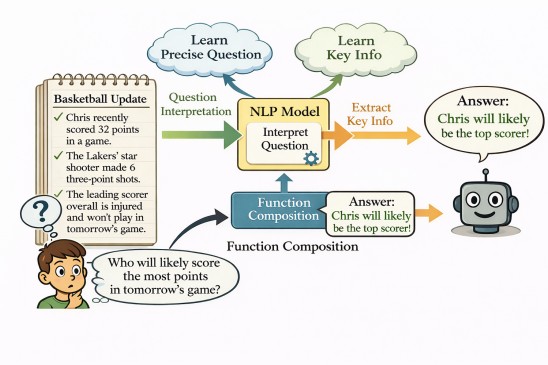

*Figure 1.* Example S2FC task. The answer to a learned question only depends on a part of the long context input.

$M(\vec{\mathbf{x}}) = F_N \circ F_{N-1} \circ \cdots \circ F_1(\vec{\mathbf{x}})$, where $\circ$ is function composition and each $F_i$ is a sequence-to-sequence map called a *layer*. We will consider types of layers.

**Transformer Layer.** Consider an embedded input sequence $\vec{\mathbf{x}} = (\mathbf{x}_1, \ldots, \mathbf{x}_L)$ such that $\mathbf{x}_i \in \mathbb{R}^d$. An attention head Attn is defined by matrices $\mathbf{W}_k, \mathbf{W}_q, \mathbf{W}_v \in \mathbb{R}^{d \times d}$ such that $\text{Attn}(\vec{\mathbf{x}})_j = \sum_{i=1}^n \alpha_{ji} \mathbf{W}_v \mathbf{x}_i$, where

$$\alpha_{ji} := \frac{\exp\left((\mathbf{W}_q \mathbf{x}_j) \cdot (\mathbf{W}_k \mathbf{x}_i)\right)}{\sum_{i=1}^n \exp\left((\mathbf{W}_q \mathbf{x}_j) \cdot (\mathbf{W}_k \mathbf{x}_i)\right)}.$$

An attention layer AT is defined by $H$ attention heads $\text{Attn}_1, \ldots, \text{Attn}_H$ and a projection matrix $\mathbf{W}_o \in \mathbb{R}^{d \times dH}$. Denote by $\mathbf{O} := (\text{Attn}_1(\vec{\mathbf{x}})^\top, \ldots, \text{Attn}_H(\vec{\mathbf{x}})^\top)^\top \in \mathbb{R}^{dH \times L}$ the concatenation of the outputs of the $H$ attention heads, so the attention layer $\text{AT}(\vec{\mathbf{x}})$ outputs $\mathbf{W}_o \mathbf{O} \in \mathbb{R}^{d \times L}$.

A Transformer layer TF is defined by an attention layer AT and an MLP layer. In particular, an MLP layer is defined as $f(\mathbf{x}) : \mathbb{R}^d \to \mathbb{R}^d$ as $f(\mathbf{x}) = \mathbf{U}_2 \sigma(\mathbf{U}_1 \mathbf{x})$, where $\mathbf{U}_1, \mathbf{U}_2$ are matrices and $\sigma$ is an activation function applied coordinate-wise. Specifically, $\text{TF}(\vec{\mathbf{x}}) = \text{MLP}(\text{AT}(\vec{\mathbf{x}}))$.

**State-Space Model Layer.** We use a similar formalism of a SSM layer as in Jelassi et al. (2024). A state space $\mathcal{S}$ is a finite set. We denote by $\text{mem}(\mathcal{S})$ the number of bits required to encode the states of $\mathcal{S}$, namely $\text{mem}(\mathcal{S}) = \log(|\mathcal{S}|)$. A *generalized state space model* (GSSM) is a layer defined by an update rule $u : \mathcal{S} \times \mathcal{V} \to \mathcal{S}$ and an output function $r : \mathcal{S} \to \mathcal{V}$. Let $s_0 \in \mathcal{S}$ be some initial state. Given some sequence $\mathbf{x}_1, \ldots, \mathbf{x}_L$, the state of the model at iteration $i$ is denoted by $s_i = S_i(\mathbf{x}_1, \ldots, \mathbf{x}_i)$ and the output token is denoted by $r_i = R_i(\mathbf{x}_1, \ldots, \mathbf{x}_i)$. The state and output are defined recursively:

1. $S_0(\emptyset) = s_0$,
2. $S_i(\mathbf{x}_1, \ldots, \mathbf{x}_i) = u(S_{i-1}(\mathbf{x}_1, \ldots, \mathbf{x}_{i-1}), \mathbf{x}_i)$,
3. $R_i(\mathbf{x}_1, \ldots, \mathbf{x}_i) = r(S_i(\mathbf{x}_1, \ldots, \mathbf{x}_i))$.

**Memory Budget.** In this work, we will compare the behavior of different models according to their memory budget. In particular, we will consider two types of budgets:

input-dependent memory and input-independent memory. *Input-dependent memory*, also called *working memory*, is the size of the intermediate state of the model, applicable to SSMs. *Input-independent memory* is used to characterize the number of parameters in the model.

# 3 Function Compositions and Limitations of Pure Models

In this work, we define the following family of tasks that we term *structured two-layer function composition (S2FC)* tasks, which expose the limitations of pure models.

**Definition 3.1** (Structured Two-Layer Function Composition (S2FC)). Let $\mathcal{V}$ be a vocabulary of tokens. Let $m, n \in \mathbb{Z}^+$. Consider a function $F(u, v) : \mathcal{V}^m \times \mathcal{V}^n \to \mathcal{Y}$. Let $u(\vec{x}) : \mathcal{V}^L \to \mathcal{V}^m$ and $v(\vec{x}) : \mathcal{V}^L \to \mathcal{V}^n$ be two functions that map a long sequence of tokens to parameters needed for computing $F$. The goal for model $M$ is to compute $M(\vec{x}) = F(u(\vec{x}), v(\vec{x}))$.

Computing deep sequential function compositions has been used as a technique for understanding the limitations of Transformer-based models empirically or through communication complexity (Chen et al., 2024; Dziri et al., 2023). Though the family of tasks we consider here shares a similar flavor to prior works, we need only consider a function of composition with depth 1. Intuitively, it is convenient to think about $u(\vec{x})$ as a subsequence of $\vec{x}$ that contains essential information that one should look at (of length $m$ for $m \ll L$ but moderately long, i.e., the width of the necessary context), while $v(\vec{x})$ can be thought as a small parameter that controls the result of $F(u, v)$.

Many *long context tasks* naturally fall into such function composition categories. For example, in a natural question answering task, the input context is usually very long, but the question (which must be learned from the context) only sparsely depends on part of the context. Transformers often struggle retrieving the information without consuming almost the whole sequence into memory, while after retrieving information, a pure SSM requires an extremely large state space to perform the rest of the computation. We start by showing that for a broad class of very simple $u, v$, *pure Transformers and pure state space models cannot compute $F(u, v)$ without sufficient scale*.

## 3.1 Limitations of SSMs

To make the above intuition formal, we first provide conditions under which $F$ is hard to compute by an SSM.

**Assumption 3.2.** Consider any function $F$ that satisfies Definition 3.1. We say the function $F$ is *hard to compute by an SSM* if it satisfies the following property: There exists a set $Q = \{v^{(i)}\}_{i=1}^q \subseteq \mathcal{V}^n$ such that $G(u) := (F(u, v^{(1)}), \ldots, F(u, v^{(q)}))$ is an injection.

Our first result shows that if $F$ satisfies Assumption 3.2, then a $k$-layer SSM either requires $k$ to be $\Omega(m)$ or needs to have one layer with number of states exponential in $m$. That is, to compute $F$, *the size of an SSM must grow linearly with respect to the hidden parameter $m$.* Formally,

**Theorem 3.3.** *Let $F$ be a function defined as in Definition 3.1 that satisfies Assumption 3.2. There is a distribution $D$ over the input $(u, v)$ such that any model $M$ that is a composition of $k$ state space layers $\mathrm{SSM}_i$, with state space $\mathcal{S}_i, i \in [k]$ that can compute $F$ with probability $1/2$ must satisfy $\sum_{i=1}^{k} \log(|\mathcal{S}_i|) \geq \Omega(m \log(|\mathcal{V}|) - q \log(|\mathcal{Y}|))$.*

*Remark* 3.4. The distribution $D$ considered in Theorem 3.3 is defined over $(u, v)$ instead of the actual distribution over the input context $\vec{x}$. For concrete tasks that satisfy Assumption 3.2, we construct distributions over $\vec{x}$ to simulate $D$.

To prove Theorem 3.3, we first prove a structural result for a pure multi-layer SSM. Roughly speaking, if a model is a sequence of multiple layers of state space models, then we can view them as single-layer SSMs. We defer the proof of Lemma 3.5 to Section C.1.

**Lemma 3.5.** *Consider a $k$-layer model $M$ defined that is a composition of $k$ state space layers $\mathrm{SSM}_j, j \in [k]$. There is a model $\mathrm{SSM}'$ that only consists of a single layer SSM, which behaves the same as $M$. In particular, denote the state space of $\mathrm{SSM}_j$ as $\mathcal{S}_j, j \in [k]$, and denote by the state space of $\mathrm{SSM}'$ as $\mathcal{S}'$, then $|\mathcal{S}'| \leq \prod_{j=1}^{k} |\mathcal{S}_j|$.*

Given Lemma 3.5 and Yao's min-max principle, we only need to consider deterministic single-layer SSMs. The main technical difficulty of the proof is that, unlike in Jelassi et al. (2024); Zhan et al. (2025), which prove hardness against specific tasks, we have little knowledge of the structure of $F$ and cannot compute the probability of failure directly. We use information theoretic arguments: at a high level, for a fixed sample prefix and $m$ different random control parameters $v$, there need to be $\Omega(m \log |\mathcal{V}|)$ bits to store all of the necessary information from the prefix to use $v$ correctly. The full proof of Theorem 3.3 is in Section C.2.

## 3.2 Limitations of Transformers

Next we study the limitations of using a Transformer to solve this problem under a memory constraint. We consider *sliding window attention*, a dominant design choice for large context models. The size of the sliding window characterizes the working memory of a Transformer based model. Our hardness result is developed in Assumption 3.6 and Theorem 3.7. Roughly speaking, when the underlying function $F$ is *globally sensitive*, which implies predicting a token at a position requires information very far from the current position, any pure Transformer model must either be very deep or have one layer that is very dense. We remark that Assumption 3.6 is very natural. In the context of S2FC,

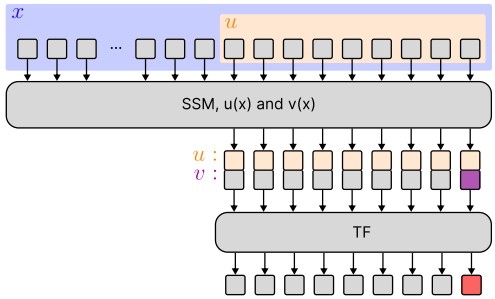

*Figure 2.* The construction's style follows taking an input $x$ and implementing 2 functions $u, v$ with an SSM. Typically, $u$ is a truncation of the input, and $v$ is a control parameter (represented in purple). Lastly, a Transformer combines these by implementing $F$ to perform the complete task (represented in red).

$F(u, v)$, though the length of the essential context $u, v$ may be small, the control parameter $v$ might be sensitive and depend on a long range of the input context, making using a standard Transformer costly. For example, if $v(\vec{x})$ is the last token in the sequence that satisfies some property, then any Transformer must maintain a very long window size in order to compute the control parameter. The proof of Theorem 3.7 is in Section C.3.

**Assumption 3.6.** Let $F$ be a function that satisfies Definition 3.1. We say the function $F(u(\vec{x}), v(\vec{x}))$ is hard to compute by a Transformer if it is $R$-globally sensitive. That is, there are two sequences $\vec{x}, \vec{x}'$ such that $\vec{x}_{L-R+1:L} = \vec{x}'_{L-R+1:L}$ but $F(\vec{x}) \neq F(\vec{x}')$.

**Theorem 3.7.** *Let $F$ be a function that satisfies Assumption 3.6. There is a distribution $D$ over the input such that any model $M$ that is a composition of $k$ Transformer layers $\mathrm{TF}_1, \ldots, \mathrm{TF}_k$ that can compute $F$ with probability $2/3$ must satisfy $\sum_{i=1}^{k} W_i \geq R$.*

## 4 Hybrid Model for S2FC

Using our hardness results, if a function $F$ satisfies Assumption 3.2 and Assumption 3.6 simultaneously, then any pure SSM must need a large number of parameters to solve the problem, making a key bottleneck for training and deploying the model, while any pure Transformer must need a working memory nearly in $\Omega(L)$ due to data redundant, making inference over long-context data a key bottleneck.

To achieve the best of both worlds, i.e., a model with a small number of parameters and relatively small working memory, we consider hybrid models that combine state-space and Transformer layers. The intuitive motivation for hybrid use is that *a state space model can implicitly act as an encoder that summarizes the information from the long context $\vec{x}$ and passes the compressed information to a Transformer*. Since the Transformer itself does not have the ability to select the correct positions to look at unless it has a deep depth or a

large window size, using more comprehensive information can reduce the space requirement of the Transformer.

To avoid making the notation messy, we think of $\vec{v}(\vec{x})$ as a single token so that given $(\vec{u}, \vec{v})$, a Transformer solves the problem by answering a query $q_v$. Suppose we have a state space model $\mathrm{SSM}_u$ parameterized by $(S^u, R^u)$ that maps $\vec{x} \in \mathcal{V}^L$ to a sequence $\vec{a} \in \mathcal{V}^L$ such that $\vec{a}_{L-m+1:L} = \vec{u}(\vec{x})$ and an encoder based model $\mathrm{SSM}_v$ parameterized by $S^v, R^v$ that maps $\vec{x} \in \mathcal{V}^L$ to a sequence $\vec{b} \in \mathcal{V}^L$ such that $\vec{b}_L = \vec{v}(\vec{x})$. We can build a merged SSM by combining the two state space models in a black box way. That is,

$$S(\mathbf{x}_1, \ldots, \mathbf{x}_i) = (S^u(\mathbf{x}_1, \ldots, \mathbf{x}_i), S^v(\mathbf{x}_1, \ldots, \mathbf{x}_i)),$$

$$R(S(\mathbf{x}_1, \ldots, \mathbf{x}_i)) = \begin{pmatrix} R^u(S^u(\mathbf{x}_1, \ldots, \mathbf{x}_i)) \\ R^v(S^v(\mathbf{x}_1, \ldots, \mathbf{x}_i)) \end{pmatrix}.$$

In matrix form, $\mathrm{SSM}$ maps $\vec{x}$ to the following matrix

$$\begin{pmatrix} r_1^u & r_2^u & , \ldots, & r_L^u \\ r_1^v & r_2^v & , \ldots, & r_L^v \end{pmatrix}.$$

In particular, if we look at the last $m$ columns of the output of the state-space model, then we have

$$\begin{pmatrix} u_1 & u_2 & , \ldots, & u_m \\ r_{L-m+1}^v & r_{L-m+2}^v & , \ldots, & v \end{pmatrix}.$$

Suppose we have a Transformer TF parameterized by $(\mathbf{W}_q^{\mathrm{TF}}, \mathbf{W}_k^{\mathrm{TF}}, \mathbf{W}_v^{\mathrm{TF}})$ such that given $(\vec{u}, \vec{v})$, it can compute $F(\vec{u}, \vec{v})$. We consider the following attention layer $\mathbf{W}_q(r_i^u, r_i^v)^\top = \mathbf{W}_q^{\mathrm{TF}} r_i^v, \mathbf{W}_k(r_i^u, r_i^v)^\top = \mathbf{W}_k^{\mathrm{TF}} r_i^u, \mathbf{W}_v(r_i^u, r_i^v)^\top = \mathbf{W}_v^{\mathrm{TF}} r_i^u$. The output of TF $\circ$ SSM is exactly $F(\vec{u}, \vec{v})$. Furthermore, such a construction *preserves the model size and the working memory of both state space models and Transformers*. We next use this idea to show that for several natural synthetic tasks, we can construct small scale hybrid models achieving good working memory efficiency.

**Selective Copying.** Our first task is defined as:

**Definition 4.1** (Selective Copying). Consider a vocabulary of tokens $\mathcal{V} = \mathcal{N} \cup \mathcal{M}$, where $\mathcal{N} = \{\#1, \#2, \ldots, \#N\}$, $|\mathcal{N}| = N$ and $|\mathcal{M}| = M$. The other values in $\mathcal{M}$ are arbitrary. Provided some sequence $(\mathbf{x}_i)_{i=1}^L$, let $i = \mathrm{argmax}_{1 \leq i \leq L} \mathbf{x}_i \in \mathcal{N}$, the goal is to extract the token $\mathbf{x}_{L+1-\mathbf{x}_i}$, i.e. $\mathbf{x}_{L+1} = \mathbf{x}_{L+1-\mathbf{x}_i}$.

As a direct application of Theorem 3.3 and Theorem 3.7, the size of a pure SSM that can solve the selective copying task well must scale linearly with respect to $|\mathcal{N}|$, while a pure Transformer must have working memory that scales linearly in the context length $L$. The proof is in Section D.1.

**Theorem 4.2.** *Consider the task of selective copying. There is a distribution $D$ over $\mathcal{V}^L$ such that any pure state space model that can solve the task for some $\vec{x}$ drawn from $D$ with*

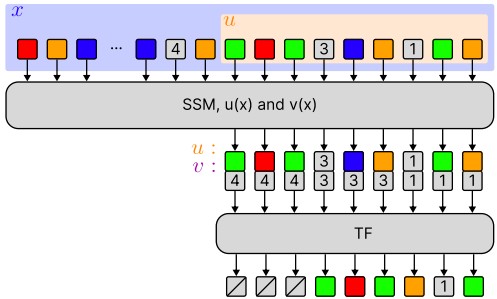

*Figure 3.* The construction solving selective copy takes an input sequence and finds the most recent number token (as represented in the bottom squares of the output of the SSM). The Transformer can then use these to look back some relative distance to find the correct token to output.

*probability* $90\%$ *must have* $\sum_{i=1}^{k} \log(|\mathcal{S}_i|) \geq N \log M$, *where* $\mathcal{S}_i$ *is the state space of the $i$th SSM layer. Furthermore, any pure Transformer model that can solve the task with probability* $90\%$ *must have* $\sum_{i=1}^{k} W_i \geq \Omega(L)$.

**Hybrid Model for Selective Copying.** To sidestep the limitations of pure models, we **design a two-layer hybrid that provably solves the problem with small input-independent memory *and* input-dependent memory**. Our construction follows the discussion in Section 4. In particular, the SSM component can be realized by a Mamba model.

**Theorem 4.3.** *There is a two-layer hybrid with a Mamba layer and an attention layer that can solve the selective copying task for* every *input sequence* $\vec{\mathbf{x}} \in \mathcal{V}^L$. *Furthermore, the hybrid model has an embedding dimension* $d = O(\max(\log|\mathcal{V}|, \log L))$ *such that the Mamba layer has* $O(|\mathcal{V}|)$ *state space, while the attention layer has dimension* $d$ *and a sliding window size of* $O(N)$.

The proof of Theorem 4.3 is in Section D.2. We remark that the number of parameters of our hybrid model is only poly $\log(\max(|\mathcal{V}|, L))$ with working memory $\tilde{O}(N)$, which is much smaller than $L$, unless $N$ is extremely large. The construction follows a structure where the SSM stores the most recent number token into its state, adding it to the current token. The Transformer can then use this information to copy the token that many positions in the past (Fig. 3).

**Associative Recall with Decoding.** As a further concrete application of our framework, we introduce another task, for which a hybrid model outperforms both pure state space models and Transformers.

**Definition 4.4** (Associative Recall with Decoding). Consider a vocabulary of tokens $\mathcal{V} = \mathcal{M} \cup \{0, 1\}$, where $\mathcal{M}$ is a set of word tokens. Let $\vec{\mathbf{x}} \in \mathcal{V}^L$ be a sequence of input tokens and let $v(\vec{x}) \in \{0, 1\}^{\log(|\mathcal{V}|)}$ be the $0 - 1$ subsequence of $\vec{\mathbf{x}}$. Denote by $\Phi(\vec{\mathbf{x}}) \in \mathcal{M}$, the token with binary representation $\vec{\mathbf{x}}$. Given $v(\vec{\mathbf{x}})$, the goal is to output the next token

in $\vec{\mathbf{x}}$ behind the last $\Phi(v(\vec{\mathbf{x}}))$ token.

As another implication of Theorem 3.3 and Theorem 3.7, the size of a pure SSM that can solve the associative recall with decoding task well must scale linearly with respect to the number of all possible word tokens, while a pure Transformer must have working memory that scales linearly in the context length $L$. The proof of Theorem 4.5 is in Section D.3.

**Theorem 4.5.** *Consider the task of associate recall with decoding. There is a distribution $D$ over $\mathcal{V}^L$ such that any pure state space model that can solve the task for some $\vec{\mathbf{x}}$ drawn from $D$ with probability $90\%$ must have $\sum_{i=1}^{k} \log(|\mathcal{S}_i|) \geq \Omega(W \log W)$, where $\mathcal{S}_i$ is the state space of the $i$th SSM layer and $W = |\mathcal{M}|$. Any pure Transformer model that can solve the task with probability $90\%$ must have $\sum_{i=1}^{k} W_i \geq \Omega(L)$.*

We next show that for very natural distributions (including the one stated in Theorem 4.5), the performance of a hybrid model can be much better than that of a pure model. In particular, we show that we can construct a hybrid model such that the number of parameters of the model scales with the logarithm of the task size. The intuition here is that for associative recall, the only tokens that matter tend to appear at the end of the sequence, which implies that the window used by the Transformer can be improved to much smaller than $L$. For example, if each token is sampled uniformly from the vocabulary, then with probability $99\%$, within a window of size $\tilde{O}(|\mathcal{M}|)$, we can see all distinct tokens from the vocabulary. In fact, the hard distribution we considered in Theorem 4.5 also satisfies such a property. Thus, once we use a state space model to extract the control variable $\vec{x}$, we are able to solve the problem with a small model with a small working memory. We defer the proof of Theorem 4.6 to Section D.4.

**Theorem 4.6.** *Consider the task of associative recall with decoding. There is a three-layer hybrid model that is a combination of a Mamba layer and two attention layers that can solve the selective copying task with probability $99\%$ for an input sequence $\vec{\mathbf{x}} \in \mathcal{V}^L$ such that tokens in $\vec{\mathbf{x}} \setminus v(\mathbf{x})$ are drawn from a uniform distribution. The hybrid model has an embedding dimension $d = O(\max(\log|\mathcal{V}|, \log L))$ such that the Mamba layer has $O(|\mathcal{V}|)$ state spaces, while the attention layer has dimension $d$ and a sliding window size of $\tilde{O}(|\mathcal{V}|)$.*

## 5   Experiments

Our theoretical results show fundamental expressivity differences between pure models and hybrid constructions. We empirically validate three claims related to these results:

- **C1.** The construction for the hybrid model empirically outperforms the pure Transformer and pure SSM base-

lines, as predicted by our theoretical results,

- **C2.** Under standard training approaches, a *learned* (rather than constructed) hybrid also outperforms pure Transformer and pure SSM baselines,
- **C3.** In further and more realistic settings, including out-of-distribution and length generalization scenarios, learned hybrids continue to outperform pure Transformer and pure SSM baselines.

Here, C1 verifies our basic theoretical claims, while C2 and C3 show that the benefits of hybrids persist in typical scenarios (e.g., the hybrid model is trained in standard ways, the tasks and settings deviate from the exact ones we studied). By validating these claims, ***we demonstrate that the fundamental benefits of hybrid models carry over to practical settings***—and are not just a theoretical curiosity.

### 5.1 Construction Implementations

Each of the constructions for Theorem 4.3 and Theorem 4.6 have been implemented to test their validity. Both perform their respective tasks on the last position in the context, as desired, validating C1. For details, see Appendix D.5.

### 5.2 Learnability Experiments

To validate C2, we conduct experiments designed to test the capacity of a hybrid to learn the two main tasks we studied, selective copying and associative recall with decoding. Besides these, we also empirically study two additional tasks: multi-key associative recall (MKAR) and needle-in-a-haystack (NH). These tasks also fall into the category of S2FC and are standard tasks used to evaluate pure Transformer-based and SSM models. Our goal is to determine whether the hybrid models learned with standard training approaches (rather than explicitly constructed) continue to outperform non-hybrids. As we shall see, our findings confirm that this is the case.

To simplify comparisons, rather than directly comparing the state sizes or the input-independent memory for the Transformers, we compare models with similar parameter counts, controlled through the embedding dimension of the tokens. Both the state size and the input-independent memory scale similarly as embedding dimension is increased.

**Experiment Details.** These models are comprised of GPT-NeoX and Mamba layers for attention and SSM layers. We use RoPE positional encodings. Unless otherwise specified, we are using windowed, causal attention, the transformers have a single head, and the Mamba state dimension expansion is 1. All experiments are trained to convergence using linearly decaying learning rates. The experiments are seq-to-seq, and accuracy if measured over all valid tokens. Transformers use a windowed attention mask.

When listed in figures, layers are read left-to-right. For

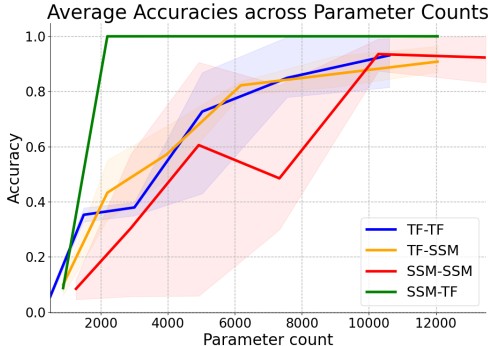

| Parameters | Pure TF | Pure SSM | TF→SSM | *SSM→TF* |
|---|---|---|---|---|
| ∼ 1000 | 0.056 | 0.084 | 0.100 | 0.087 |
| ∼ 2000 | 0.352 | 0.305 | 0.433 | 0.999 |
| ∼ 6000 | 0.727 | 0.485 | 0.822 | 1.000 |
| ∼ 12000 | 0.923 | 0.931 | 0.908 | 1.000 |

*Figure 4.* Results from training small models on Selective Copy, across an increase in the hidden dimension of the models. At 2000 parameters, hybrid models consistently attain perfect accuracy. The pure models, with 6x the parameters, only attain around 0.9 accuracy.

example, SSM-TF is an SSM layer followed by a TF layer.

**Selective Copy.** *Setup.* Given the relative simplicity of this task, we can study model expressivity for small models, with up to approximately ten thousand parameters.

*Results.* We depict the results in Figure 4. As expected, the hybrid performs the task to 90% accuracy with significantly fewer parameters than either the pure Transformer or pure SSM, by around a factor of $6\times$. Also note that the "reverse" hybrid, with the Transformer layer first, performs no differently than the pure models. This is consistent with prior work (Park et al., 2024).

**Associative Recall with Decoding.** *Setup.* For this task, we expect both the pure Transformer and the pure SSM to struggle, while a hybrid has the expressive power to represent this task. Unlike the other tasks, this experiment utilized three layer models rather than two layer ones. This is due to the construction for Theorem 4.6, where three layers were needed. These models required significantly more parameters than the other models, close to 1 million.

*Results.* For these three layer models, we observe the same behavior: the hybrid excels while the pure models struggle. As seen in Figure 5, at the scales tested, none of the pure models achieved greater than 40% accuracy, while the hybrid did at much smaller scales, eventually surpassing 50% accuracy. At a high level, this task is more difficult than the others as it requires the more complex computation of binary values before performing the commonly tested task of associative recall.

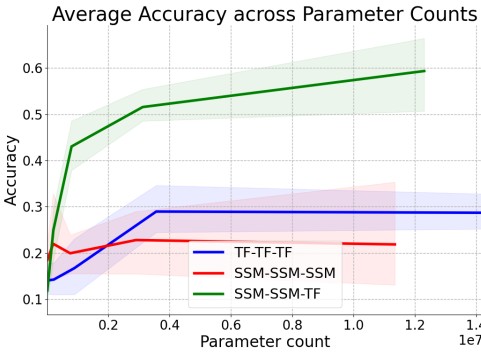

*Figure 5.* Results from training small models on Associative Recall with Decoding. Even at much smaller scales than the pure models, the hybrid is the only architecture that attains 0.5 accuracy. At the scales tested, none of the pure models performed the task with more than 0.4 accuracy.

**Multi-Key Associative Recall.** *Setup.* We now turn our attention to a further pair of tasks, the first being MKAR.

**Definition 5.1** (MKAR). Let $\vec{x}$ be a sequence of length $L$ sampled from a vocabulary $\mathcal{V}$, and let $k$ be some small number. Let $K = \vec{x}_{L-k:L}$. Let $i$ be the last position in the context where $\vec{x}_{i:i+k} = K$. Performing *MKAR* is outputting $\vec{x}_{i+k}$.

In the framework of S2FC, we can take $v$ to be the empty map, $u$ to include enough context to find the key, and $F$ to be the look-up operation. Since $F$ can have any output depending heavily on the context, this is hard for an SSM. Since $u$ could be large, this is also hard for a Transformer. However, since $v$ is empty, S2FC does not immediately indicate if a separation exists between Transformers and hybrids.

*Results.* In Figure 6, we see that SSMs perform quite poorly, while hybrids and Transformers can perform the task at scale. However, we see a similar separation between hybrids and Transformers present for selective copying. Specifically, hybrids perform the task on average with $6\times$ fewer parameters than the pure Transformers to an accuracy of $60\%$.

**Needle in a Haystack.** *Setup.* To complement MKAR, we trained the same models on needle-in-a-haystack (NH).

**Definition 5.2** (NH). Let $\vec{x}$ be a sequence of length $L$ sampled from a vocabulary $\mathcal{V} \cup \{M\}$. Let the location of $M$ be denoted by $i^*$. Performing *NH* is outputting $\vec{x}_{i^*+1}$.

This task is simple: copy the token(s) after a marker token when requested for at the end of the input sequence. This task is hard for Transformers due to windowing, while SSMs and hybrids should perform this task easily. In the framework of S2FC, $u$ is the empty map, and $F$ is the identity. As such, we do not have the hardness criterion for SSMs; we should therefore not immediately expect hybrids to perform

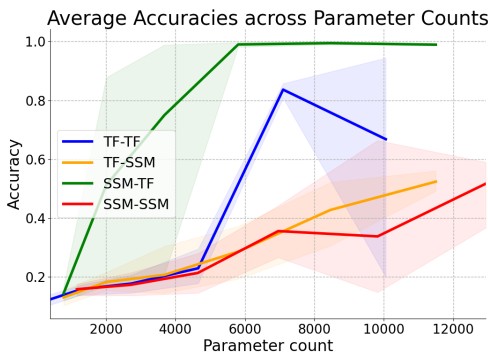

| Parameters | Pure TF | Pure SSM | TF→SSM | *SSM→TF* |
|---|---|---|---|---|
| $\sim 1000$ | 0.124 | 0.158 | 0.131 | 0.144 |
| $\sim 2000$ | 0.159 | 0.173 | 0.183 | 0.512 |
| $\sim 6000$ | 0.230 | 0.356 | 0.286 | 0.990 |
| $\sim 12000$ | 0.668 | 0.517 | 0.524 | 0.989 |

*Figure 6.* Results from training small models on Multi-Key Associative Recall, across an increase in the hidden dimension. The hybrid consistently outperforms the pure models of the same depth and similar parameter counts. The hybrid models could perform the task to 60% accuracy with $6\times$ fewer parameters than any of the pure Transformers.

differently than SSMs.

*Results.* We show results in Figure 7 for full-context attention. Even in the situation where the Transformer could possibly learn the task, small token dimensions result in learnability issues. SSMs also perform more inconsistently than hybrid models in small parameter regimes. The mechanism behind these separations is not directly characterized by S2FC and is left to future work.

For both of these tasks, we see the same property: on these synthetic tasks and at the scales tested, **hybrids outperform pure models, even when we use tasks that are outside of the S2FC framework.**

### 5.3 Further Experiments

In contrast to the much smaller expressivity experiments, our next set of experiments are designed to increase the scale of the models to somewhere nearer those of modern LLMs. These models will have around 100 million parameters each, closer to the scale of standard language models. Different properties of these models are tested since accuracy on many of the above tasks already approaches 1, and scaling up difficulty parameters, such as vocabulary size, which only increases the difficulty by a small amount, leads to models with similar behaviors to the expressivity experiments.

**Associative Recall with Decoding.** To analyze these different analyses, we use the more difficult task of associative recall with decoding. Empirically, this task proved far more challenging than selective copy, where 2-layer models with

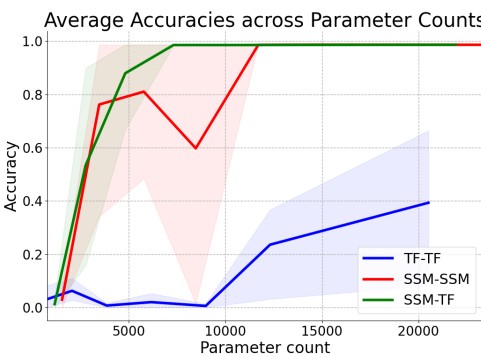

*Figure 7.* Results from training small models on Needle in a Haystack, across an increase in the hidden dimension of the models with no context windowing. The hybrid and SSM perform this task with fewer parameters than the Transformer, however we still see the hybrid with a slight improvement. This task was expected to be hard for the Transformer and not the SSM.

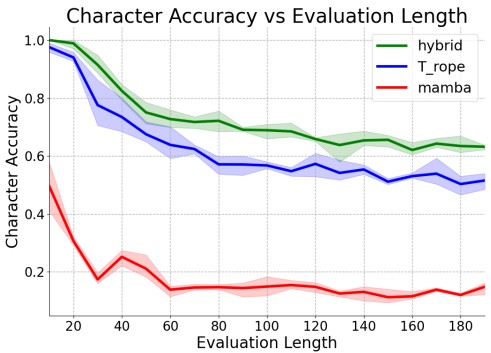

*Figure 8.* The distribution of accuracies across different input sequence lengths. Hybrid models with comparatively similar parameters as their attention/Transformer counterparts perform better at longer lengths consistently.

the same scale of parameters learned nothing.

**Length Generalization.** *Setup.* First, we investigate the length generalization of different models. Each model is trained on sequences of length 20 to 50, and tests on longer sequences as well. Comparing the hybrid model to the Transformer (T_rope) and SSM (mamba), Figure 8 shows how the hybrid models *consistently* outperforms the pure models.

*Results.* As expected, the performance drops as sequences grow longer; however, hybrids lose performance at the slowest rate. This means that even though hybrids and Transformers behave within 2% of each other for short sequences, this separation grows to be around 10% for longer sequences.

**OOD Generalization.** *Setup.* We also tested these models as their sampling distributions are changed. Specifically, we tested Associative Recall with Decoding with differing proportions of bits between test and train time. This task

| Train Proportion | SSM | TF | Hybrid |
|:---:|:---:|:---:|:---:|
| 0.05 | 0.24 | **0.47** | **0.47** |
| 0.1 | 0.34 | 0.40 | **0.47** |
| 0.3 | 0.17 | 0.64 | **0.74** |
| 0.5 | 0.46 | 0.63 | **0.77** |
| 0.8 | 0.67 | 0.63 | **0.83** |
| 0.9 | **0.86** | 0.61 | 0.80 |

*Table 1.* Results from training 12-layer models with different proportions of bits for Associative Recall with Decoding. Data are evaluation accuracies for evaluation bit proportions of 0.2. Each architecture tends to improve performance as the training bit proportion increases, with hybrids consistently out-performing the pure models.

allows us to test these behaviors on larger models, as the other two tasks saturate, only showing 100% accuracy on all tests. Under these distributions, we can see which of the architectures learns representations that consistently perform well across different varying sampling distributions.

*Results.* The results can be seen in Table 1. For almost all training distributions, the hybrid indeed performs the best on a 0.2 proportion test set. However, there are some other notable trends within these results beyond just the hybrid's performance. In particular, the different architectures show varying behavior across training distributions. SSMs tend to improve the most as more training bits are added, while Transformers improve the least. Hybrid models attain the best of both works, acting well with both a high frequency and a low frequency of bits.

## 6 Conclusion

We studied when hybrid sequence models combining SSM and attention layers can simultaneously achieve strong expressivity and favorable memory scaling. We formalized S2FC tasks that require both (i) extracting a control variable from long context and (ii) performing content-addressable retrieval conditioned on that variable. Under natural conditions, we showed that pure SSMs and sliding-window Transformers each face fundamental memory limitations on this family. In contrast, we constructed small hybrids that provably solve selective copying and associative recall with decoding while using substantially smaller working memory than either pure counterpart. Experiments on learned models corroborated these separations, showing that hybrids can outperform larger pure baselines and generalize better to longer sequences and distribution shifts. Limitations include our focus on synthetic tasks and restricted Transformer attention mechanisms; extending the theory to broader attention patterns, external memory, identifying real function-composition datasets, and natural long-context workloads are important directions for future work.

# 7 Acknowledgments

John Cooper and Frederic Sala were supported by the National Science Foundation (NSF) (CCF2106707), the Defense Advanced Research Projects Agency (DARPA Young Faculty Award) and the Wisconsin Alumni Research Foundation (WARF). Ilias Diakonikolas was supported by the H.I. Romnes Faculty Fellowship, and the ONR (N00014-25-1-2268). Mingchen Ma was supported by NSF Award CCF-2144298 (CAREER).

## Impact Statement

This paper presents work whose goal is to advance the field of Machine Learning. There are many potential societal consequences of our work, none which we feel must be specifically highlighted here.

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

## Supplementary Materials

We provide supplementary materials here. In Section A, we provide related works. In Section B, we give a complete list of preliminaries and notations. In Section C, we present omitted proofs in Section 3, establishing hardness results for SSMs and transformers. In Section D, we provide omitted proofs in Section 4, providing concrete hybrid model constructions. In Section E, we provide additional details to our empirical experiments.

## A   Related Works

**State Space Models.** State space models (SSMs) are a classical framework (Elman, 1990; Hochreiter & Schmidhuber, 1997) for modeling sequential data via the evolution of a latent state governed by a dynamical system. Recently, SSMs have attracted renewed interest in machine learning as efficient alternatives to attention-based architectures for long-sequence modeling, owing to their linear-time inference and favorable memory scaling. In their canonical form, SSMs describe sequences through linear state transitions and observation maps, providing a principled mechanism for compressing historical information into a fixed-dimensional representation. This revival includes structured SSM architectures that carefully design or learn the state dynamics to capture long-range dependencies, such as HiPPO (Gu et al., 2020) and S4(Gu et al., 2022b), as well as simplified variants like S4D(Gu et al., 2022a). More recently, selective and input-dependent SSMs—most notably Mamba (Gu & Dao, 2024), have demonstrated strong empirical performance at scale.

**Hybrid Architectures.** Hybrid sequence models that combine state space/recurrent dynamics with attention mechanisms have gained increasing attention as a means to balance efficient long-range memory with expressive contextual interaction. In the Transformer era, models like Transformer-XL (Dai et al., 2019) explored recurrent memory within attention-based architectures, providing early empirical evidence for hybrid designs. With the modern revival of structured state space models, works such as (Fu et al., 2022; Arora et al., 2023), which combine SSM layers with a few attention layers, have shown that hybrids can match or exceed Transformer performance on language tasks while retaining linear-time context dynamics. These findings attracted recent interest in empirical studies of hybrid models on in-context learning (Park et al., 2024), language tasks (Lee et al., 2025) and large-scale empirical validation (Lieber et al., 2024; Ren et al., 2024). A consistent theme exists for each of these studies: empirically, hybrid models tend to perform better than pure models of similar sizes, especially on long-sequence tasks. However, many miss a rigorous understanding of what the structure of these tasks are which requires a mixture of these pure models.

**Expressive Power and Efficiency Tradeoffs.** The expressive power of pure state space models and pure Transformers has been extensively studied through the lenses of computational and communication complexity (Merrill & Sabharwal, 2023; Peng et al., 2024; Merrill et al., 2024; Chen et al., 2024; Yehudai et al., 2025), providing purely theoretical characterizations of the classes of problems each architecture can solve. Complementary to this line of work, a growing body of research—motivated by the challenges of long-context reasoning (Waleffe et al., 2024)—investigates expressivity and memory–computation efficiency using carefully designed synthetic tasks in controlled empirical settings. These tasks are designed to probe specific capabilities such as long-range copying, indexing, and associative retrieval, including repeat-copy tasks (Jelassi et al., 2024) and associative recall benchmarks (Fu et al., 2022; Arora et al., 2023). For example, (Jelassi et al., 2024) shows that state space models have difficulty copying long contexts, whereas a small Transformer can solve the same task under mild assumptions. More recently, (Zhan et al., 2025) introduces joint recall tasks that are provably hard for pure state space models in sub-quadratic time, yet become tractable when a state space model is augmented with a *context-dependent* sparse attention layer. Despite these advances, the fundamental tradeoffs between expressive power and efficiency for hybrid architectures remain poorly understood.

## B   Complete Preliminaries and Notations

We consider sequence to sequence token prediction problem. Let $\mathcal{V}$ be some vocabulary of tokens and $V = |\mathcal{V}|$ and $\vec{\mathbf{x}} = (\mathbf{x}_i)_{i=1}^L$ be and input sequence. A language model $M$ is sequence to sequence map $M : \mathcal{V}^L \to \mathcal{V}^m$ of the form $M(\vec{\mathbf{x}}) = F_N \circ F_{N-1} \circ \cdots \circ F_1(\vec{\mathbf{x}})$, where each $F_i$ is a sequence to sequence map called layer.

We will consider several different layers in this paper.

**Transformer Layer.** Consider an input $\mathbf{x}_1, \ldots, \mathbf{x}_L$ such that $\mathbf{x}_i \in \mathbb{R}^d$. An attention head $\mathrm{Attn}$ is defined by matrices

| Symbol | Meaning |
|--------|---------|
| $\phi, \psi, \Phi$ | Token and Positional Embeddings |
| $u, v$ | Control parameters of the task |
| $F$ | The target task |
| $H$ | (Relative) Entropy |
| $I$ | Mutual Information |
| $\mathbf{W}_q, \mathbf{W}_k, \mathbf{W}_v, \mathbf{W}_o$ | Transformer parameters |
| $\mathbf{W}_A, \mathbf{W}_B, \mathbf{W}_C, \Delta$ | SSM parameters |
| $\mathbf{x}$ | The input sequence |
| $\mathcal{V}$ | Vocabulary space |
| $\mathcal{N}, \mathcal{M}$ | Number/Vocabulary components of $\mathcal{V}$ |
| $Y$ | Target space, typically $\mathcal{V}^n$ |
| $d$ | The token dimension |
| $d_s$ | The state dimension |

*Table 2.* Notation.

$\mathbf{W}_k, \mathbf{W}_q, \mathbf{W}_v \in \mathbb{R}^{d \times d}$ such that $\text{Attn}(\vec{\mathbf{x}})_j = \sum_{i=1}^{n} \alpha_{ji} \mathbf{W}_v \mathbf{x}_i$, where

$$\alpha_{ji} := \frac{\exp\left((\mathbf{W}_q \mathbf{x}_j) \cdot (\mathbf{W}_k \mathbf{x}_i)\right)}{\sum_{i=1}^{n} \exp\left((\mathbf{W}_q \mathbf{x}_j) \cdot (\mathbf{W}_k \mathbf{x}_i)\right)}.$$

*Remark* B.1. We remark that for some practical applications, a bias term $B$ will also be added to the computation of attention.

An attention layer AT is defined by $H$ attention heads $\text{Attn}_1, \ldots, \text{Attn}_H$ and a projection matrix $\mathbf{W}_o \in \mathbb{R}^{d \times dH}$. Denote by $\mathbf{O} := (\text{Attn}_1(\vec{\mathbf{x}})^{\top}, \ldots, \text{Attn}_H(\vec{\mathbf{x}})^{\top})^{\top} \in \mathbb{R}^{dH \times L}$ the concatenation of the outputs of the $H$ attention heads, so the attention layer $\text{AT}(\vec{\mathbf{x}})$ outputs $\mathbf{W}_o \mathbf{O} \in \mathbb{R}^{d \times L}$.

A Transformer layer TF is defined by an attention layer AT and an MLP layer MLP. In particular, an MLP layer is defined as $f(\mathbf{x}) : \mathbb{R}^d \to \mathbb{R}^d$ as $f(\mathbf{x}) = \mathbf{U}_2 \sigma(\mathbf{U}_1 \mathbf{x})$, where $\mathbf{U}_1, \mathbf{U}_2$ are matrices and $\sigma$ is an activation function applied coordinate-wise. Specifically, $\text{TF}(\vec{\mathbf{x}}) = \text{MLP}(\text{AT}(\vec{\mathbf{x}}))$.

**State Space Model Layer.** We use a similar definition of SSM layer as (Jelassi et al., 2024). A state space $\mathcal{S}$ is some finite set. We denote by $\text{mem}(\mathcal{S})$ the number of bits required to encode the states of $\mathcal{S}$, namely $\text{mem}(\mathcal{S}) = \log(|\mathcal{S}|)$. A *generalized state space layer* (GSSM) is a sequence model defined by an update rule $u : \mathcal{S} \times \mathcal{V} \to \mathcal{S}$ and some output function $r : \mathcal{S} \to \mathcal{V}$. Let $s_0 \in \mathcal{S}$ be some initial state. Given some sequence $\mathbf{x}_1, \ldots, \mathbf{x}_L$, the state of the model at iteration $i$ is denoted by $S_i(\mathbf{x}_1, \ldots, \mathbf{x}_i)$ and the output token is denoted by $R_i(\mathbf{x}_1, \ldots, \mathbf{x}_i)$. The state and output are defined recursively:

1. $S_0(\emptyset) = s_0$,

2. $S_i(\mathbf{x}_1, \ldots, \mathbf{x}_i) = u(S_{i-1}(\mathbf{x}_1, \ldots, \mathbf{x}_{i-1}), \mathbf{x}_i)$,

3. $R_i(\mathbf{x}_1, \ldots, \mathbf{x}_i) = r(S_i(\mathbf{x}_1, \ldots, \mathbf{x}_i))$.

**Mamba Layer.** Our SSM layers in our constructions are defined as follows. Let $\text{Mamba}^{(i)}$ be a Mamba layer. Let $d$ be the token embedding dimension size and $d_s$ be the state dimension size. Let

$$\mathbf{W}_A \in \mathbb{R}^{d_s \times d_s}, \mathbf{W}_B \in \mathbb{R}^{d_s \times d}, \mathbf{W}_C \in \mathbb{R}^{d \times d_s}, \Delta(\mathbf{x}_t) \in \mathbb{R}$$

be constants and a function that returns the shift in time $\Delta(\mathbf{x}_t)$ based on the current token. We update the hidden state matrix as follows:

$$H_t = (I - \Delta(\mathbf{x}_t)\mathbf{W}_A)H_{t-1} + \Delta(\mathbf{x}_t)\mathbf{W}_B \mathbf{x}_t$$
$$y_t = \mathbf{W}_C H_t$$

This follows from the first-order approximation of the S6, where $\exp(-\Delta A) \approx I - \Delta A$. This is useful for construction purposes to avoid large constants $M \gg 1$ to push $\exp$ close to 0. Equivalent constructions can be found by instead placing a large constant $M$ is specific locations with the original $\exp$.

We also omit the typical per-token dependencies that exist for $\mathbf{W}_B$ and $\mathbf{W}_C$ in standard Mamba, as constant functions were sufficient for our constructions.

In this work, we will consider two kinds of models used in practice, encoder-based models and decoder-based models. We next formally defined the two models as follows.

**Encoder-Based Model.** An Encoder-Based Model $M$ can be thought of as a sequence-to-sequence map. That is to say, given a sequence of input tokens $\vec{x} = (\mathbf{x}_i)_{i=1}^L$, the model $M$ maps it to another sequence $\vec{y} = (y_i)_{i=1}^L$, each token $y_i$ corresponds to a token $\mathbf{x}_i$ in the sequence. Specifically, let $M = F_N \circ F_{N-1} \circ \cdots \circ F_1(\vec{x})$ be a language model with $N$ layers. Each layer $F_t$ has an input $h^{(t-1)} \in \mathbb{R}^{d \times L}$ and an output $h^{(t)} \in \mathbb{R}^{d \times L}$, in particular $h^{(0)}$ is the embedding of the input sequence $\vec{x}$. Furthermore, for every $t \in [N]$ and $i \in [L]$, $h_i^{(t)}$ is a function of the whole vector $h^{(t-1)}$.

**Decoder-Based Model(Autoregressive Model).** A Decoder-Based Model $M$ can be thought of as an autoregressive(generative) model. Roughly speaking, for each input sequence $\vec{x}$, we consider recursively generating $\mathbf{x}_{L+i} = M(\mathbf{x}_1, \ldots, \mathbf{x}_{L+i-1})$ and denote by $M(\vec{x}) = \mathbf{x}_{L+1}, \mathbf{x}_{L+2}, \ldots$ the final output of the model $M$. For a multi-layer model $M = F_N \circ F_{N-1} \circ \cdots \circ F_1(\vec{x})$, the inference stage of $M$ contains two stages. In the first stage, each layer $F_t$ has an input $h^{(t-1)} \in \mathbb{R}^{d \times L}$ and an output $h^{(t)} \in \mathbb{R}^{d \times t}$, in particular $h^{(0)}$ is the embedding of the input sequence $\vec{x}$. However, unless the encoder-based model, $h_i^{(t)}$ only depends on $h_1^{(t-1)}, \ldots, h_i^{(t-1)}$. We will decode $h_L^{(N)}$ as $\mathbf{x}_{L+1}$ and enter the second stage, where we generate $\mathbf{x}_{L+2}$ by consuming $(\mathbf{x}_1, \ldots, \mathbf{x}_{L+1})$.

**Memory Budget.** In this work, we will compare the behavior of different models according to their memory budget. In particular, we will consider two types of budgets: input-dependent memory and input-independent memory. By input-dependent memory, we mean the space needed for storing the input as well as the intermediate results. By input-independent memory, we mean the number of parameters of the model.

**Embeddings.** A binary embedding for a finite space $\mathcal{X}$ will be a map $\psi'_{d'} : \mathcal{X} \to \{-1, 1\}^{d'}$ where $d' \geq \log_2 |\mathcal{X}|$ or larger.

## C   Omitted Proofs in Section 3

### C.1   Proof of Lemma 3.5

**Lemma C.1** (Restatement of Lemma 3.5). *Consider a $k$-layer auto-regressive SSM $M$ defined as $\mathrm{SSM}_1 \to \mathrm{SSM}_2 \to \cdots \to \mathrm{SSM}_k$, where for $j \in [k]$, $\mathrm{SSM}_j$ is an SSM layer. There is a model $\mathrm{SSM}'$ that only consists a single layer SSM, which behaves the same as $M$. In particular, denote the state space of $\mathrm{SSM}_j$ as $\mathcal{S}_j$, $j \in [k]$, and denote by the state space of $\mathrm{SSM}'$ as $\mathcal{S}'$, then $|\mathcal{S}'| \leq \prod_{j=1}^k |\mathcal{S}_j|$.*

*Proof of Lemma 3.5.* Let $s_0^{(j)}, u^{(j)}, r^{(j)}, R_i^{(j)}$, and $S_i^{(j)}$ be the SSM parameters that define $\mathrm{SSM}_j$. The output of this $k$-layer model is computed as follows:

$$S_0^{(j)}(\emptyset) = s_0^{(j)} \quad \text{for } j \in [k]$$

$$S_i^{(1)}(\mathbf{x}_1, \ldots, \mathbf{x}_i) = u^{(1)}(S_{i-1}^{(1)}(\mathbf{x}_1, \ldots, \mathbf{x}_{i-1}), \mathbf{x}_1)$$

$$S_i^{(j)}(R_1^{(j-1)}, \ldots, R_i^{(j-1)}) = u^{(j)}(S_{i-1}^{(j)}(R_1^{(j-1)}, \ldots, R_{i-1}^{(j-1)}), R_i^{(j-1)}) \quad \text{for } 2 \leq j \leq k$$

$$R_i^{(j)}(R_1^{(j-1)}, \ldots, R_i^{(j-1)}) = r^{(j)}(S_i^{(j)}(R_1^{(j-1)}, \ldots, R_i^{(j-1)}))$$

The output of each layer is fed into the next layer to update its state. We define $\mathcal{S}'$ to be $\{(s_1, \ldots, s_k) \mid s_i \in \mathcal{S}^i, i \in [k]\}$, so $|\mathcal{S}'| = \prod_{i=1}^k |\mathcal{S}^i|$. Let $\mathrm{SSM}'$ have state

$$S_i'(\mathbf{x}_1, \ldots, \mathbf{x}_i) = \left( S_i^{(1)}(\mathbf{x}_1, \ldots, \mathbf{x}_i),\ S_i^{(2)}(R_1^{(1)}, \ldots, R_i^{(1)}),\ \ldots,\ S_i^{(j)}(R_1^{(j-1)}, \ldots, R_i^{(j-1)}) \right).$$

We remark that the definition above is well defined, since for every $j \in [k]$, $(R_1^{(j)}, \ldots, R_i^j)$ only depends on $(\mathbf{x}_1, \ldots, \mathbf{x}_i)$, thus is available at time $i$. The output function $r'$ is defined to be the output of the final layer of the multilayer SSM $(S_i')_k$,

$$r'(S_i'(\mathbf{x}_1, \ldots, \mathbf{x}_i)) = R_i^{(k)}((S_i')_k)$$

All other parameters are simply their application across the different components of the compound state.

$$S_0'(\emptyset) = (s_0^{(1)}, \ldots, s_0^{(j)})$$

$$u'(S_{i-1}'(\mathbf{x}_1, \ldots, \mathbf{x}_{i-1}), \mathbf{x}_i) = (S_i^{(1)}, \ldots, S_i^{(j)})$$

This construction computes the same result as the original multilayer model with the same size of state. □

## C.2 Proof of Theorem 3.3

**Theorem C.2** (Restatement of Theorem 3.3). *Let $F$ be a function defined Definition 3.1 that satisfies Assumption 3.2. There is a distribution $D$ over the input $(u, v)$ such that any model $M$ that is a composition of $k$ state space layers $\mathrm{SSM}_i$, with state space $\mathcal{S}_i, i \in [k]$ that can compute $F$ with probability $1/2$ must satisfy $\sum_{i=1}^{k} \log(|\mathcal{S}_i|) \geq \Omega(m \log(|\mathcal{V}|) - q \log(|\mathcal{Y}|))$.*

*Proof of Theorem 3.3.* By Lemma 3.5, we only need to show the hardness against a single layer of state space model. By Yao's min-max principle, it is sufficient to show that we cannot construct any deterministic state-space model that can solve the problem with a good probability when the input is drawn from some distribution $D$. We consider the following distribution $D$ over a sequence of length $m + n$ tokens. For the first $m$ tokens, we sample each token uniformly from $\mathcal{V}$, representing $u$ is drawn uniformly at random. For the last $n$ tokens, we will draw $v \sim Q$ independent on $u$. We next lower bound the size of $|\mathcal{S}|$ of any SSM that can give the correct output when the input $(u, v)$ is drawn from $D$. Denote by $Y_i = F(u, v^{(i)})$ for $i \in [q]$ and $Y = (Y_1, \ldots, Y_q)$. Furthermore, let $s = S_m(u)$ the random state of the model after reading $u$. By Assumption 3.2, we know that $H(s \mid u) = 0$. Since

$$I(u; s) = H(u) - H(u \mid s) = H(s) - H(s \mid u),$$

we have

$$H(u) - H(u \mid s) = H(s) \leq \log(|\mathcal{S}|),$$

because the support of $s$ has size at most $|\mathcal{S}|$. We next upper bound $H(u \mid s)$. By the symmetry of mutual information, we have

$$H(u \mid s) = H(Y \mid s) + H(u \mid Y, s) - H(Y \mid u, s)$$

$$= H(Y \mid s) \leq \sum_{i=1}^{q} H(Y_i \mid s),$$

where the second equation holds by $H(u \mid Y, s) - H(Y \mid u, s) = 0$. It remains to upper bound $H(Y_i \mid s)$. Let $\mathrm{err}_i := \mathbf{Pr}_{u \sim \mathcal{V}^m}(R(S_{m+n}(u, v^{(i)})) \neq Y_i)$. By Fano's inequality, we have

$$H(Y_i \mid s) \leq H_2(\mathrm{err}_i) + \mathrm{err}_i \log(|\mathcal{Y}|).$$

Thus,

$$H(u \mid s) \leq q\frac{1}{q} \sum_{i=1}^{q} H(Y_i \mid s)$$

$$\leq q\frac{1}{q} \sum_{i=1}^{q} (H_2(\mathrm{err}_i) + \mathrm{err}_i \log(|\mathcal{Y}|))$$

$$\leq q(H_2(\mathrm{err}) + \mathrm{err} \log(|\mathcal{Y}|))$$

This implies, if we set up $\mathrm{err} < 1/8$, then we have

$$\log(|\mathcal{S}|) \geq m \log(|\mathcal{V}|) - q(H_2(1/8) + \log(|\mathcal{Y}|)/8).$$

To conclude the proof of Theorem 3.3, we discuss the possible choice of $q$ and $|\mathcal{Y}|$ that satisfy Assumption 3.2. Since $G(u) := (F(u, v^{(1)}), \ldots, F(u, v^{(q)}))$ is an injection, we know that the smallest possible choice of $q, |\mathcal{Y}|$ satisfies $q \log(|\mathcal{V}|) = O(m \log(|\mathcal{V}|))$. This implies $\log(|\mathcal{S}|) \geq \Omega(m \log(|\mathcal{V}|))$

□

## C.3 Proof of Theorem 3.7

**Theorem C.3** (Restatement of Theorem 3.7). *Let $F$ be a function that satisfies Assumption 3.6. There is a distribution $D$ over the input such that any model $M$ that is a composition of $k$ Transformer blocks $\mathrm{TF}_1, \ldots, \mathrm{TF}_k$ that can compute $F$ with probability $2/3$ must satisfy $\sum_{i=1}^{k} W_i \geq R$.*

*Proof of Theorem 3.7.* Consider any sliding window Transformer $M$ with window size $W_i$ for the $i$-th layer. Denote by $W = \sum_{i=1}^{k} W_i$ the effective window size of a Transformer. Notice that the output of $M$ is a deterministic function of $(\mathbf{x}_{L-W+1}, \ldots, \mathbf{x}_L)$. Thus, we construct a distribution $D$ that draws $\vec{x}$ and $\vec{x'}$ uniformly. For any input drawn from $D$, $M$ has the same output. However, by definition, $F(\vec{x}) \neq F(\vec{x'})$, which implies that $M$ fails to output correctly with probability $1/2$.

$\square$

# D   Omitted Proofs in Section 4

## D.1   Proof of Theorem 4.2

**Theorem D.1** (Restatement of Theorem 4.2). *Consider the task of selective copying. There is a distribution $D$ over $\mathcal{V}^L$ such that any pure state space model that can solve the task for some $\vec{x}$ drawn from $D$ with probability $90\%$ must have $\sum_{i=1}^{k} \log(|\mathcal{S}_i|) \geq N \log M$, where $\mathcal{S}_i$ is the state space of the $i$th SSM layer, furthermore, any pure Transformer model that can solve the task with probability $90\%$ must have $\sum_{i=1}^{k} W_i \geq \Omega(L)$.*

*Proof of Theorem 4.2.* We write down the task in the form of a function composition $F(u(\vec{x}), v(\vec{x}))$. Let $u(\vec{x}) = \vec{x}_{L-N+1:L}$ and $v(\vec{x}) := \mathrm{argmax}_{1 \leq i \leq L} x_i \in \mathcal{N}$ and $F(u, v)$ be $u_{L+1-v}$.

We first show that $F(u, v)$ satisfies Assumption 3.2. We take $v^{(i)} = i, i \in [N]$, which implies $F(u, v^{(i)}) = u_i, i \in [N]$. Thus, $G(u) = (F(u, v^{(1)}), \ldots, F(u, v^{(N)})) = u$ is an injection. Taking $m = q = N$, by Theorem 3.3, we know that when a pure SSM gives a sequence of input of the form $(u, v)$, where $u$ is drawn uniformly from $\mathcal{V}$ and $v$ is drawn uniformly from $\mathcal{N}$, then if the SSM wants to solve the task with probability at least $7/8$, it needs

$$\sum_{i=1}^{k} \log(|\mathcal{S}_i|) \geq \Omega(N \log(|M|)),$$

when a pure SSM is given $(u, v)$ such that $u$ is uniformly drawn from $\mathcal{V}^m$ and $v$ is drawn uniformly from $\mathcal{N}$. To simulate such a distribution over $(u, v)$ using a long context $\vec{x}$, we select the first $L - 1$ tokens from $\mathcal{V}$ uniformly at random, while selecting the last token as a random token from $\{2, \ldots, N\}$. Notice that such a distribution $D_S$ is sufficient to simulate the required distribution of $(u, v)$, since the effective part of $u$ is $u_{1:N-1}$ and $u$ is $u_{1:N-1}$ is independent on $v$.

We next show that $F(u(x), v(x))$ is $L/2$ sensitive. Let $\vec{x}, \vec{x'} \in \mathcal{V}^L$ be any input context such that $\vec{x}_{L/2:L} = \vec{x'}_{L/2:L}$ and $\vec{x}_i \notin \mathcal{N}$, for every $i = L/2, \ldots, L$. This implies $v(\vec{x})$ and $v(\vec{x'})$ only depends on the first $L/2$ coordinates. Thus, $F(u(\vec{x}), v(\vec{x}))$ is $L/2$ sensitive. By Theorem 3.7, we know that any pure Transformer model that can compute $F(u(\vec{x}), v(\vec{x}))$ with a constant probability must have $\sum_{i=1}^{k} W_i \geq \Omega(L)$. In particular, to construct a distribution $D_T$ over $\vec{x}$ that makes a pure Transformer fail, we draw $\vec{x}$ such that $\vec{x}_{L/2:L}$ is chosen uniformly from $\mathcal{M}$ and $\vec{x}_{L/2:L}$ is chosen uniformly from $\mathcal{V}$.

To conclude the proof of Theorem 4.2, we will choose a distribution $D := D_S/2 + D_T/2$. Under this distribution, a pure SSM with $\sum_{i=1}^{k} \log(|\mathcal{S}_i|) < \Omega(N \log(|M|))$ has a constant failure probability when $\vec{x}$ is drawn from $D_S$ and a Transformer with working memory less than $o(L)$ has a constant failure probability when $\vec{x}$ drawn from $D_T$. $\square$

## D.2   Proof of Theorem 4.3

**Theorem D.2** (Restatement of Theorem 4.3). *Consider the task of selective copying. There is a two-layer hybrid model that is a combination of a mamba layer and an attention layer that can solve the selective copying task for* every *input sequence $\vec{x} \in \mathcal{V}^L$. Furthermore, the hybrid model has an embedding dimension $d = O(\max(\log |\mathcal{V}|, \log L))$ such that the Mamba layer has $O(|\mathcal{V}|)$ state spaces, while the attention layer has dimension $d$ and a sliding window size of $O(N)$.*

*Proof of Theorem 4.3.* We first construct the two-layer hybrid model and show the correctness of the construction.

**Token Embeddings.** We begin by constructing an embedding function that maps each $\mathbf{x} \in \mathcal{V}$ to a vector in $\mathbb{R}^{d'}$ for some $d > 0$. Let $d' = \log |\mathcal{V}|$. We define $\psi' : \mathcal{V} \to \{\pm 1\}^{d'}$ to be the binary encoding of the vocabulary $\mathcal{V}$. For each $\mathbf{x} \in \mathcal{V}$, we embed the token $\mathbf{x}$ as

$$\psi(\mathbf{x}) = \begin{pmatrix} \psi'(\mathbf{x}) & \mathbb{1}_{\{\mathbf{x} \in \mathcal{N}\}} \psi'(\mathbf{x}) & \mathbf{0} \end{pmatrix}^\top.$$

Here $\mathbf{0} \in \mathbb{R}^\ell$ for some $\ell \leq O(\log(|\mathcal{V}|) + \log L)$ is a zero vector. For a given input context $\vec{\mathbf{x}}$, the embedded context has the form of

$$\begin{pmatrix} \psi'(\mathbf{x}_1) & \psi'(\mathbf{x}_2) & \dots & \psi'(\mathbf{x}_{L-1}) & \psi'(\mathbf{x}_L) \\ \mathbb{1}_{\{\mathbf{x}_1 \in \mathcal{N}\}} \psi'(\mathbf{x}_1) & \mathbb{1}_{\{\mathbf{x}_2 \in \mathcal{N}\}} \psi'(\mathbf{x}_2) & \dots & \mathbb{1}_{\{\mathbf{x}_{L-1} \in \mathcal{N}\}} \psi'(\mathbf{x}_{L-1}) & \mathbb{1}_{\{\mathbf{x}_L \in \mathcal{N}\}} \psi'(\mathbf{x}_L) \\ \mathbf{0} & \mathbf{0} & \dots & \mathbf{0} & \mathbf{0} \end{pmatrix}.$$

**Position Encoding.** To allow the attention layer to access the position of the input tokens, we next add a position encoding to each input token. Define $\phi' : [L] \to \{\pm 1\}^{\log L}$ be the binary encoding of numbers in $L$. For each $i \in [L]$, we encode the position $i$ using a function $\phi$ defined as follows

$$\phi(i) = \phi'(L + 1 - i).$$

After adding the position encoding, the input context has the following form.

$$\Phi(\vec{\mathbf{x}}) := \begin{pmatrix} \psi'(\mathbf{x}_1) & \dots & \psi'(\mathbf{x}_{L-1}) & \psi'(\mathbf{x}_L) \\ \mathbb{1}_{\{\mathbf{x}_1 \in \mathcal{N}\}} \psi'(\mathbf{x}_1) & \dots & \mathbb{1}_{\{\mathbf{x}_{L-1} \in \mathcal{N}\}} \psi'(\mathbf{x}_{L-1}) & \mathbb{1}_{\{\mathbf{x}_L \in \mathcal{N}\}} \psi'(\mathbf{x}_L) \\ \mathbf{0} & \dots & \mathbf{0} & \mathbf{0} \\ \phi(1) & \dots & \phi(L-1) & \phi(L) \end{pmatrix}$$

By construction, each column of the input context is in $\mathbb{R}^d$, for some $d = O(\log(|\mathcal{V}| + \log L)$. Given the embedding of the input context, we now construct the SSM layer and the attention layer of the hybrid model.

**Mamba Layer.** We define the weights of the Mamba layer as follows. Let $\mathbf{W}_A, \mathbf{W}_B, \mathbf{W}_C$ be as follows,

$$\mathbf{W}_B \begin{pmatrix} \psi'(\mathbf{x}_i) \\ \mathbb{1}_{\{\mathbf{x}_i \in \mathcal{N}\}} \psi'(\mathbf{x}_i) \\ \mathbf{0} \\ \phi'(i) \end{pmatrix} = \mathbb{1}_{\{\mathbf{x}_i \in \mathcal{N}\}} \psi'(\mathbf{x}_i), \quad \mathbf{W}_C \mathbf{v} = \begin{pmatrix} \mathbf{0} \\ \mathbf{0} \\ \mathbf{v} \\ \mathbf{0} \end{pmatrix},$$

$\mathbf{W}_A = I$, and $\Delta(\mathbf{x}) = \mathbb{1}_{\{\mathbf{x} \in \mathcal{N}\}}$. We claim the following guarantee for the constructed Mamba layer.

*Claim* D.3. Let $\vec{\mathbf{x}} \in \mathcal{V}^L$ be a sequence of input contexts. Given the embedded context $\Phi(\vec{\mathbf{x}})$, the SSM layer with parameter $\mathbf{W}_A, \mathbf{W}_B, \mathbf{W}_C, \Delta(\mathbf{x})$, has the following output

$$\begin{pmatrix} \psi'(\mathbf{x}_1) & \psi'(\mathbf{x}_2) & \dots & \psi'(\mathbf{x}_{L-1}) & \psi'(\mathbf{x}_L) \\ \mathbb{1}_{\{\mathbf{x}_1 \in \mathcal{N}\}} \psi'(\mathbf{x}_1) & \mathbb{1}_{\{\mathbf{x}_2 \in \mathcal{N}\}} \psi'(\mathbf{x}_2) & \dots & \mathbb{1}_{\{\mathbf{x}_{L-1} \in \mathcal{N}\}} \psi'(\mathbf{x}_{L-1}) & \mathbb{1}_{\{\mathbf{x}_L \in \mathcal{N}\}} \psi'(\mathbf{x}_L) \\ \phi(L+1-n_1) & \phi(L+1-n_2) & \dots & \phi(L+1-n_{L-1}) & \phi(L+1-n_L) \\ \mathbf{0} & \mathbf{0} & \dots & \mathbf{0} & \mathbf{0} \\ \phi(1) & \phi(2) & \dots & \phi(L-1) & \phi(L) \end{pmatrix},$$

where $n_i = \mathbf{x}_{\mathrm{argmax}_{1 \leq j \leq i} \mathbf{x}_j \in \mathcal{N}}$

*Proof of Claim D.7.* By our choice of $\Delta(\mathbf{x}_t)$, if $\mathbf{x}_t \in \mathcal{N}$, then $H_t = \mathbf{W}_B \Phi(\mathbf{x}_t)$ and if $\mathbf{x}_t \notin \mathcal{N}$, then $H_t = H_{t-1}$. Notice that if we set $H_0$ to be a zero vector, then by induction, for each $t$, $H_t$ is a sparse vector, with the only non-zero component $\phi(L + 1 - n_i) = \psi'(n_i)$. Using an MLP layer to combine the output with the input, we know that the input context

$$\begin{pmatrix} \psi'(\mathbf{x}_1) & \psi'(\mathbf{x}_2) & \dots & \psi'(\mathbf{x}_{L-1}) & \psi'(\mathbf{x}_L) \\ \mathbb{1}_{\{\mathbf{x}_1 \in \mathcal{N}\}} \psi'(\mathbf{x}_1) & \mathbb{1}_{\{\mathbf{x}_2 \in \mathcal{N}\}} \psi'(\mathbf{x}_2) & \dots & \mathbb{1}_{\{\mathbf{x}_{L-1} \in \mathcal{N}\}} \psi'(\mathbf{x}_{L-1}) & \mathbb{1}_{\{\mathbf{x}_L \in \mathcal{N}\}} \psi'(\mathbf{x}_L) \\ \phi(L+1-n_1) & \phi(L+1-n_2) & \dots & \phi(L+1-n_{L-1}) & \phi(L+1-n_L) \\ \mathbf{0} & \mathbf{0} & \dots & \mathbf{0} & \mathbf{0} \\ \phi(1) & \phi(2) & \dots & \phi(L-1) & \phi(L) \end{pmatrix}.$$

$\square$

**Attention Layer.** Based on the output of the Mamba layer, we will now construct an attention layer that can solve the copy task. Let $\mathbf{W}_q, \mathbf{W}_k, \mathbf{W}_v$ be the weight matrices of the attention layer.

$$\mathbf{W}_q \begin{pmatrix} \psi'(\mathbf{x}_i) \\ \mathbb{1}_{\{\mathbf{x}_i \in \mathcal{N}\}} \psi'(L+1-\mathbf{x}_i) \\ \phi'(n_i) \\ \mathbf{0} \\ \phi'(i) \end{pmatrix} = M\phi(n_i), \quad \mathbf{W}_k \begin{pmatrix} \psi'(\mathbf{x}_i) \\ \mathbb{1}_{\{\mathbf{x}_i \in \mathcal{N}\}} \psi'(L+1-\mathbf{x}_i) \\ \phi'(n_i) \\ \mathbf{0} \\ \phi'(i) \end{pmatrix} = \phi(i)$$

$$\mathbf{W}_v \begin{pmatrix} \psi'(\mathbf{x}_i) \\ \mathbb{1}_{\{\mathbf{x}_i \in \mathcal{N}\}} \psi'(L+1-\mathbf{x}_i) \\ \phi'(n_i) \\ \mathbf{0} \\ \phi'(i) \end{pmatrix} = \begin{pmatrix} \mathbf{0} \\ \mathbf{0} \\ \mathbf{0} \\ \psi'(\mathbf{x}_i) \\ \mathbf{0} \end{pmatrix}$$

We summarize the performance of the attention layer as the following claim.

*Claim* D.4. Let $\mathrm{SSM}(\vec{\mathbf{x}})$ be the output of the Mamba layer constructed above. By applying an attention mechanism with parameters $\mathbf{W}_q, \mathbf{W}_k, \mathbf{W}_v$ with a window size of $N$, the last output vector is a sparse vector with the only non-zero part $\psi(\mathbf{x}_{L+1-n_L})$.

*Proof of Claim D.4.* Notice that the last output vector is defined as

$$\sum_{i=L+1-N}^{L} \frac{\exp(M\phi(n_L)\phi(i))}{\sum_{i=L+1-N}^{L} \exp(M\phi(n_L)\phi(i))} \begin{pmatrix} \mathbf{0} \\ 0 \\ 0 \\ \psi'(\mathbf{x}_i) \\ 0 \end{pmatrix} = \begin{pmatrix} \mathbf{0} \\ 0 \\ 0 \\ \psi'(\mathbf{x}_{L+1-n_L}) \\ 0 \end{pmatrix}$$

$\square$

In fact, if we use a full attention, then after passing the attention layer, the context now has the form

$$\begin{pmatrix} \psi'(\mathbf{x}_1) & \psi'(\mathbf{x}_2) & \dots & \psi'(\mathbf{x}_{L-1}) & \psi'(\mathbf{x}_L) \\ \mathbb{1}_{\{\mathbf{x}_1 \in \mathcal{N}\}} \psi'(\mathbf{x}_1) & \mathbb{1}_{\{\mathbf{x}_2 \in \mathcal{N}\}} \psi'(\mathbf{x}_2) & \dots & \mathbb{1}_{\{\mathbf{x}_{L-1} \in \mathcal{N}\}} \psi'(\mathbf{x}_{L-1}) & \mathbb{1}_{\{\mathbf{x}_L \in \mathcal{N}\}} \psi'(\mathbf{x}_L) \\ \phi(n_1) & \phi(n_2) & \dots & \phi(n_{L-1}) & \phi(n_L) \\ \psi'(x_{L+1-n_1}) & \psi'(x_{L+1-n_2}) & \dots & \psi'(x_{L+1-n_{L-1}}) & \psi'(x_{L+1-n_L}) \\ \phi(1) & \phi(2) & \dots & \phi(L-1) & \phi(L) \end{pmatrix}.$$

This implies that by applying the natural decoding that copies only the row with $\psi'(\mathbf{x}_{L+1-n_1})$. The final sequence (up to arbitrarily small error) has the form

$$\begin{bmatrix} \psi'(\mathbf{x}_{L+1-n_1}) & \psi'(\mathbf{x}_{L+1-n_2}) & \dots & \psi'(\mathbf{x}_{L+1-n_{L-1}}) & \psi'(\mathbf{x}_{L+1-n_L}) \end{bmatrix}$$

To conclude the proof of Theorem 4.3, it remains to count the number of parameters and the working memory of the constructed hybrid model. $\square$

## D.3 Proof of Theorem 4.5

**Theorem D.5** (Restatement of Theorem 4.5). *Consider the task of associative recall with decoding. There is a distribution $D$ over $\mathcal{V}^L$ such that any pure state space model that can solve the task for some $\vec{\mathbf{x}}$ drawn from $D$ with probability $90\%$ must have $\sum_{i=1}^{k} \log(|\mathcal{S}_i|) \geq \Omega(W \log W)$, where $\mathcal{S}_i$ is the state space of the $i$th SSM layer and $W = |\mathcal{M}|$, furthermore, any pure Transformer model that can solve the task with probability $90\%$ must have $\sum_{i=1}^{k} W_i \geq \Omega(L)$.*

*Proof of Theorem 4.5.* To prove the hardness of the task, we will consider a constraint version of the problem. To do this, we partition equally partition the vocabulary $\mathcal{M}$ into $\mathcal{M}_1 = \{\alpha_1, \dots, \alpha_{W/2}\}, \mathcal{M}_2 = \{\beta_1, \dots, \beta_{W/2}\}$. For every possible

input $\vec{x}$, we restrict $\vec{x}$ of the following form. $\vec{x} = (\alpha_{i1}, \beta_{i1}, b_{i1}, \ldots, \alpha_{ik}, \beta_{ik}, b_{ik}) \in \mathcal{V}^L$. Here, for $j \in [l]$, $b_{ij}$ either does not appear or $b_{ij} \in \{0, 1\}$. Now, we partition $\vec{x}$ into two parts. Let $\vec{b} = (b_{i1}, \ldots, b_{ik})$ be the 0-1 subsequence of $\vec{x}$ and $\vec{w} = (\alpha_{i1}, \beta_{i1}, b_{i1}, \ldots, \alpha_{ik}, \beta_{ik}, b_{ik})$ be subsequence of $\vec{x}$ with every token in $\mathcal{M}$. Given this partition, we write the problem as a function composition. For each $\alpha \in \mathcal{M}_1$, let $\beta(\alpha) \in \mathcal{M}_2$ be the next token of the last appearance of $\alpha$ in $\vec{w}$. Let $u = (\beta(\alpha))_{\alpha \in \mathcal{M}_1} \in \mathcal{M}_2^k$ and let $v \in \mathcal{M}_1$ be the token corresponding to binary representation $\vec{b}$ of elements in $\mathcal{M}_1$. Then $F(u, v) = \beta(v)$. We notice that by choosing $G(u) = (\beta(\alpha_1), \ldots, \beta(\alpha_{W/2}))$ is an injection. Taking $m = q = W/2$, by Theorem 3.3, we know that when a pure SSM gives a sequence of input of the form $(u, v)$, where $u$ is uniformly drawn from $\mathcal{M}_2^{M/2}$ and $v$ is drawn uniformly from $\mathcal{M}_1$, then if the SSM wants to solve the task with probability at least $7/8$, it needs

$$\sum_{i=1}^{k} \log(|\mathcal{S}_i|) \geq \Omega(W \log(|W|)),$$

To simulate such a distribution over $(u, v)$ using a distribution $D_S$ over a long context $\vec{x}$, we select each $(\alpha_{ij}, \beta_{ij})$ uniformly at random and after selecting $\vec{w}$, we randomly select $\vec{b}$ and append it to $\vec{w}$.

On the other hand, by sampling $\vec{w}$ uniformly from, we know that with probability at least 99%, for each $\alpha \in \mathcal{M}_1$, the last appearance of $\alpha$ must be at the last $\tilde{O}(W)$ positions of $\vec{w}$. By Theorem 3.7, we know that any pure Transformer model that can compute $F(u(\vec{x}), v(\vec{x}))$ with a constant probability must have $\sum_{i=1}^{k} W_i \geq \Omega(L)$ if we randomly selecting $\vec{b}$ and appending it before $\vec{w}$. we denote the resulting distribution by $D_T$.

To conclude the proof of Theorem 4.5, we will choose a distribution $D := D_S/2 + D_T/2$. Under this distribution, a pure SSM with $\sum_{i=1}^{k} \log(|\mathcal{S}_i|) < \Omega(W \log(W))$ has a constant failure probability when $\vec{x}$ is drawn from $D_S$ and a Transformer with working memory less than $o(L)$ has a constant failure probability when $\vec{x}$ drawn from $D_T$.

$\square$

### D.4 Proof of Theorem 4.6

**Theorem D.6.** *Restatement of Theorem 4.6 Consider the task of associative recall with decoding. There is a three-layer hybrid model that is a combination of a mamba layer and two attention layers that can solve the associative recall with decoding task with probability 99% for an input sequence $\vec{x} \in \mathcal{V}^L$ drawn from a uniform distribution. Furthermore, the hybrid model has an embedding dimension $d = O(\max(\log |\mathcal{V}|, \log L))$ such that the Mamba layer has $O(|\mathcal{V}|)$ state spaces, while the attention layer has dimension $d$ and a sliding window size of $\tilde{O}(|\mathcal{V}|)$.*

*Proof of Theorem 4.6.* We first construct the three-layer hybrid model and show the correctness of the construction.

**Token Embeddings** Let $\mathcal{B} = \{0, 1\}$ be the set containing the two bits in the vocabulary. We begin by constructing an embedding function that maps each $\mathbf{x} \in \mathcal{V}$ to a vector in $\mathbb{R}^{d'}$ for some $d > 0$. Let $d' = \log |\mathcal{V}|$. We define $\psi' : \mathcal{V} \to \{\pm 1\}^{d'}$ to be the binary encoding of the vocabulary $\mathcal{V}$. For each $\mathbf{x} \in \mathcal{V}$, we embed the token $\mathbf{x}$ as

$$\psi(\mathbf{x}) = \begin{pmatrix} \psi'(\mathbf{x}) & \mathbf{0} & \mathbb{1}_{\{\mathbf{x} \in \mathcal{B}\}} \psi'(\mathbf{x}) & \mathbf{0} \end{pmatrix}^{\top}.$$

Here $\mathbf{0} \in \mathbb{R}^{\ell}$ for some $\ell \leq O(\log(|\mathcal{V}|) + \log L)$ is a zero vector. Also, $\mathbb{1}_{\{\mathbf{x} \in \mathcal{B}\}}$. For a given input context $\vec{x}$, the embedded context has the form of

$$\begin{pmatrix} \psi'(\mathbf{x}_1) & \psi'(\mathbf{x}_2) & \ldots & \psi'(\mathbf{x}_{L-1}) & \psi'(\mathbf{x}_L) \\ \mathbf{0} & \mathbf{0} & \ldots & \mathbf{0} & \mathbf{0} \\ \mathbb{1}_{\{\mathbf{x}_1 \in \mathcal{B}\}} \psi'(\mathbf{x}_1) & \mathbb{1}_{\{\mathbf{x}_2 \in \mathcal{B}\}} \psi'(\mathbf{x}_2) & \ldots & \mathbb{1}_{\{\mathbf{x}_{L-1} \in \mathcal{B}\}} \psi'(\mathbf{x}_{L-1}) & \mathbb{1}_{\{\mathbf{x}_L \in \mathcal{B}\}} \psi'(\mathbf{x}_L) \\ \mathbf{0} & \mathbf{0} & \ldots & \mathbf{0} & \mathbf{0} \end{pmatrix}.$$

**Position Encoding.** To allow the attention layer to access the position of the input tokens, we next add a position encoding to each input token. Define $\phi : [L] \to \{\pm 1\}^{\log L}$ be the binary encoding of numbers in $L$. After adding the position encoding,

the input context has the following form.

$$\Phi(\vec{\mathbf{x}}) := \begin{pmatrix} \psi'(\mathbf{x}_1) & \ldots & \psi'(\mathbf{x}_{L-1}) & \psi'(\mathbf{x}_L) \\ \mathbf{0} & \ldots & \mathbf{0} & \mathbf{0} \\ \mathbb{1}_{\{\mathbf{x}_1 \in \mathcal{B}\}} \psi'(\mathbf{x}_1) & \ldots & \mathbb{1}_{\{\mathbf{x}_{L-1} \in \mathcal{B}\}} \psi'(\mathbf{x}_{L-1}) & \mathbb{1}_{\{\mathbf{x}_L \in \mathcal{B}\}} \psi'(\mathbf{x}_L) \\ \mathbf{0} & \ldots & \mathbf{0} & \mathbf{0} \\ \phi(1) & \ldots & \phi(L-1) & \phi(L) \end{pmatrix}$$

By construction, each column of the input context is in $\mathbb{R}^d$, for some $d = O(\log(|\mathcal{V}| + \log L)$. Given the embedding of the input context, we now construct the SSM layer and the attention layer of the hybrid model.

**Mamba Layer.** We will need a state dimension $d_s$ equal to the number of bits per bit sequence. Let $\mathbf{W}_A = I - S$ be the block diagonal matrix, where $S$ is the permutation matrix satisfying

$$Sz = (z_2, \ldots, z_{d_s-1}, z_{d_s}, 0), \forall z \in \mathbb{R}^{d_s}.$$

We define $\mathbf{W}_B$ and $\mathbf{W}_C$ as follows,

$$\mathbf{W}_B \begin{pmatrix} \psi'(\mathbf{x}_i) \\ \mathbf{0} \\ \mathbb{1}_{\{\mathbf{x}_i \in \mathcal{B}\}} \psi'(\mathbf{x}_i) \\ \mathbf{0} \\ \phi'(i) \end{pmatrix} = \begin{pmatrix} \mathbf{0} \\ \mathbb{1}_{\{\mathbf{x}_i \in \mathcal{B}\}} \psi'(x_i) \end{pmatrix}, \quad \mathbf{W}_C \mathbf{v} = \begin{pmatrix} \mathbf{0} \\ \mathbf{0} \\ \mathbf{0} \\ \mathbf{v} \\ \mathbf{0} \end{pmatrix},$$

and $\Delta(\mathbf{x}) = \mathbb{1}_{\{\mathbf{x} \in \mathcal{B}\}}$. We claim the following guarantee for the constructed Mamba layer.

*Claim* D.7. Let $\vec{\mathbf{x}} \in \mathcal{V}^L$ be a sequence of input contexts. Given the embedded context $\Phi(\vec{\mathbf{x}})$, the SSM layer with parameters $\mathbf{W}_A, \mathbf{W}_B, \mathbf{W}_C, \Delta(\mathbf{x})$, has the following output

$$\begin{pmatrix} \psi'(\mathbf{x}_1) & \psi'(\mathbf{x}_2) & \ldots & \psi'(\mathbf{x}_{L-1}) & \psi'(\mathbf{x}_L) \\ \mathbf{0} & \mathbf{0} & \ldots & \mathbf{0} & \mathbf{0} \\ \mathbb{1}_{\{\mathbf{x}_1 \in \mathcal{B}\}} \psi'(\mathbf{x}_1) & \mathbb{1}_{\{\mathbf{x}_2 \in \mathcal{B}\}} \psi'(\mathbf{x}_2) & \ldots & \mathbb{1}_{\{\mathbf{x}_{L-1} \in \mathcal{B}\}} \psi'(\mathbf{x}_{L-1}) & \mathbb{1}_{\{\mathbf{x}_L \in \mathcal{B}\}} \psi'(\mathbf{x}_L) \\ \phi(n_1) & \phi(n_2) & \ldots & \phi(n_{L-1}) & \phi(n_L) \\ \mathbf{0} & \mathbf{0} & \ldots & \mathbf{0} & \mathbf{0} \\ \phi(1) & \phi(2) & \ldots & \phi(L-1) & \phi(L) \end{pmatrix},$$

where $n_i = v(\vec{\mathbf{x}}_{1:i})$ corresponds to token in $\mathcal{M}$ that matches the binary representation of the $0 - 1$ subsequence of $\vec{\mathbf{x}}_{1:i}$.

*Proof of Claim D.7.* We initialize $H_0 = 0$ and do induction over $H_t$. By our construction, only the third block of $H_t$ is non-zero. So, for the convenience of notation, we use $H_t$ to denote the third block of the state. Assuming that in time step $t$, $H_t = \phi(n_t)$, we prove this for time step $t + 1$. Write $\phi(n_t) = (z_1, \ldots, z_{d'})$. If $\mathbf{x}_{t+1} \notin \{0, 1\}$, then $\Delta(\mathbf{x}_{t+1}) = 0$, which implies $H_{t+1} = H_t$ and $n_t = n_{t+1}$. If $\mathbf{x}_{t+1} \in \{0, 1\}$, then $H_{t+1} = SH_t + \mathbf{x}_{t+1} = \phi(n_{t+1})$. Thus, the third block of the matrix is always $\phi(n_t)$. This implies, after using an MLP layer to combine the output with the input sequence, we know that in the input context has the form of

$$\begin{pmatrix} \psi'(\mathbf{x}_1) & \psi'(\mathbf{x}_2) & \ldots & \psi'(\mathbf{x}_{L-1}) & \psi'(\mathbf{x}_L) \\ \mathbf{0} & \mathbf{0} & \ldots & \mathbf{0} & \mathbf{0} \\ \mathbb{1}_{\{\mathbf{x}_1 \in \mathcal{B}\}} \psi'(\mathbf{x}_1) & \mathbb{1}_{\{\mathbf{x}_2 \in \mathcal{B}\}} \psi'(\mathbf{x}_2) & \ldots & \mathbb{1}_{\{\mathbf{x}_{L-1} \in \mathcal{B}\}} \psi'(\mathbf{x}_{L-1}) & \mathbb{1}_{\{\mathbf{x}_L \in \mathcal{B}\}} \psi'(\mathbf{x}_L) \\ \phi(n_1) & \phi(n_2) & \ldots & \phi(n_{L-1}) & \phi(n_L) \\ \mathbf{0} & \mathbf{0} & \ldots & \mathbf{0} & \mathbf{0} \\ \phi(1) & \phi(2) & \ldots & \phi(L-1) & \phi(L) \end{pmatrix},$$

$\square$

**Transformer Block.** Based on the output of the Mamba layer, we will now construct a Transformer block that can solve the recall task. The Transformer block contains two layers. The first layer contains two heads, while the second layer contains only one head.

We start with the construction of the first layer. The first layer maps each token $\mathbf{x}_t$ to $(\mathbf{x}_{t-1}, \mathbf{x}_t)^\top$. We denote by $\text{Attn}^{(1)}, \text{Attn}^{(2)}$ the two heads of the first attention layer $\text{AT}^{(1)}$. For the first head, we define $\mathbf{W}_q^{(1)} = \mathbf{W}_k^{(1)} = 0$, $(B_i)_j = -\infty \mathbb{1}(j \neq i-1)$. That is to say $\text{Attn}^{(1)}$ is used to select the previous element for each position. We set $\mathbf{W}_v^{(1)}$ such that

$$\mathbf{W}_v^{(1)} \begin{pmatrix} \psi'(\mathbf{x}_i) \\ \mathbf{0} \\ \mathbb{1}_{\{\mathbf{x}_i \in \mathcal{B}\}} \psi'(\mathbf{x}_i) \\ \mathbf{0} \\ \phi'(i) \end{pmatrix} = \begin{pmatrix} \mathbf{0} \\ \psi'(\mathbf{x}_i) \\ \mathbf{0} \\ \mathbf{0} \\ \mathbf{0} \end{pmatrix}$$

For the second head, we set $\mathbf{W}_q^{(1)} = \mathbf{W}_k^{(1)} = 0$, with $(B_i)_j = -\infty \mathbb{1}(j \neq i)$. And we set $\mathbf{W}_v^{(2)} = I$. This implies given any input sequence $\vec{\mathbf{x}}$, we have

$$\text{AT}^{(1)}(\text{SSM}(\vec{\mathbf{x}})) = \begin{pmatrix} \psi'(\mathbf{x}_1) & \psi'(\mathbf{x}_2) & \ldots & \psi'(\mathbf{x}_{L-1}) & \psi'(\mathbf{x}_L) \\ \psi'(\mathbf{x}_0) & \psi'(\mathbf{x}_1) & \ldots & \psi'(\mathbf{x}_{L-2}) & \psi'(\mathbf{x}_{L-1}) \\ \mathbb{1}_{\{\mathbf{x}_1 \in \mathcal{B}\}} \psi'(\mathbf{x}_1) & \mathbb{1}_{\{\mathbf{x}_2 \in \mathcal{B}\}} \psi'(\mathbf{x}_2) & \ldots & \mathbb{1}_{\{\mathbf{x}_{L-1} \in \mathcal{B}\}} \psi'(\mathbf{x}_{L-1}) & \mathbb{1}_{\{\mathbf{x}_L \in \mathcal{B}\}} \psi'(\mathbf{x}_L) \\ \phi(n_1) & \phi(n_2) & \ldots & \phi(n_{L-1}) & \phi(n_L) \\ \mathbf{0} & \mathbf{0} & \ldots & \mathbf{0} & \mathbf{0} \\ \phi(1) & \phi(2) & \ldots & \phi(L-1) & \phi(L) \end{pmatrix}.$$

That is to say, the first attention layer maps each column of the input into a query $\mathbf{x}_i^q = \phi(n_i)$, a key $\mathbf{x}_i^k = \psi'(\mathbf{x}_{i-1})$, and a value $\mathbf{x}_i^v = \psi'(\mathbf{x}_i)$. We next design the second layer $\text{AT}^{(2)}$ that contains only a single head to perform the attention mechanism using the output of $\text{AT}^{(1)}$. Let $\mathbf{W}_q, \mathbf{W}_k, \mathbf{W}_v$ be the weight matrices of the attention layer.

$$\mathbf{W}_q \begin{pmatrix} \psi'(\mathbf{x}_i) \\ \psi'(\mathbf{x}_{i-1}) \\ \mathbb{1}_{\{\mathbf{x}_i \in \mathcal{B}\}} \psi'(\mathbf{x}_i) \\ \phi(n_i) \\ \mathbf{0} \\ \phi(i) \end{pmatrix} = M\phi(n_i), \quad \mathbf{W}_k \begin{pmatrix} \psi'(\mathbf{x}_i) \\ \psi'(\mathbf{x}_{i-1}) \\ \mathbb{1}_{\{\mathbf{x}_i \in \mathcal{B}\}} \psi'(\mathbf{x}_i) \\ \phi(n_i) \\ \mathbf{0} \\ \phi(i) \end{pmatrix} = \psi'(\mathbf{x}_{i-1}), \quad \mathbf{W}_v \begin{pmatrix} \psi'(\mathbf{x}_i) \\ \psi'(\mathbf{x}_{i-1}) \\ \mathbb{1}_{\{\mathbf{x}_i \in \mathcal{B}\}} \psi'(\mathbf{x}_i) \\ \phi(n_i) \\ \mathbf{0} \\ \phi(i) \end{pmatrix} = \begin{pmatrix} \mathbf{0} \\ \mathbf{0} \\ \mathbf{0} \\ \mathbf{0} \\ \psi'(\mathbf{x}_i) \\ \mathbf{0} \end{pmatrix}$$

We also add a bias $B$ for each position, so that the argmax is achieved at the last recall token. This implies the last output vector is

$$\sum_{i=1}^{L} \frac{\exp(M(\phi(n_L)\phi(\mathbf{x}_{i-1}) + B_{Li}))}{\sum_{i=1}^{L} \exp((M\phi(n_L)\phi(\mathbf{x}_{i-1}) + B_{Li}))} \begin{pmatrix} \mathbf{0} \\ \mathbf{0} \\ \mathbf{0} \\ \mathbf{0} \\ \psi'(\mathbf{x}_i) \\ \mathbf{0} \end{pmatrix} = \begin{pmatrix} \mathbf{0} \\ \mathbf{0} \\ \mathbf{0} \\ \mathbf{0} \\ \psi'(\mathbf{x}_{i+1}^*) \\ \mathbf{0} \end{pmatrix}$$

This implies that the hybrid model outputs the correct recall token. We remark that when tokens in $\mathcal{M}$ are drawn uniformly, with probability 99%, each token in $\mathcal{M}$ appears among the last $\tilde{O}(W)$ tokens in $\vec{\mathbf{x}}$. This implies that instead of using a window of size $L$, a window of size $\tilde{O}(W)$ is enough to get the same output.

$\square$

## D.5 Construction Implementations

Both of these constructions are implemented in the code repository. We show here the input embedding and the output of these different constructions. Selective copying's construction can be found in Figure 9 and Associative Recall with

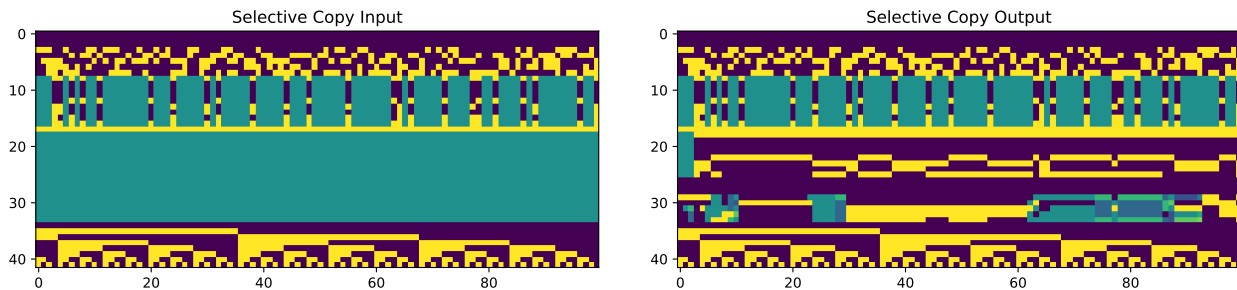

*Figure 9.* An example of the input/embedding and the output for selective copy. The aspects of the construction are kept in relatively similar positions in the implementation. Dark purple is -1, cyan is 0, and yellow is 1.

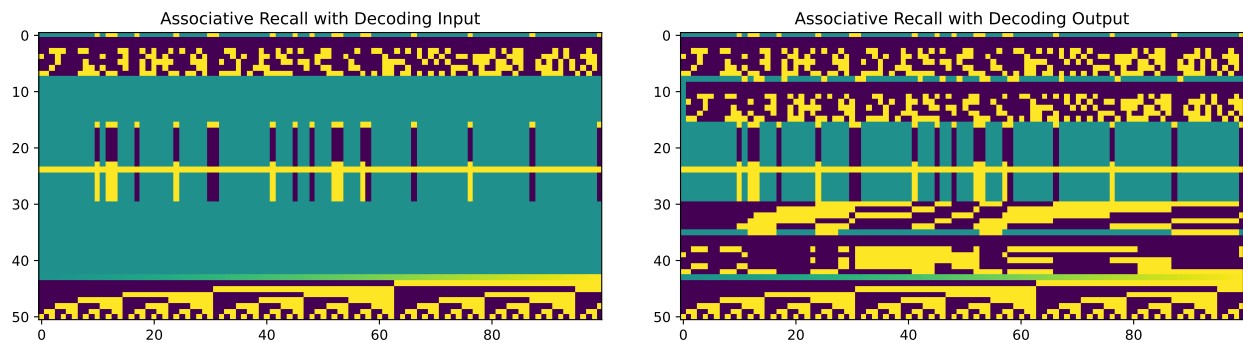

*Figure 10.* An example of the input/embedding and the output for associative recall with decoding. The aspects of the construction are kept in relatively similar positions in the implementation. Dark purple is -1, cyan is 0, and yellow is 1.

Decoding's construction can be found in Figure 10. On interesting aspect of these constructions comes from their similarity and dissimilarity to the structures in learned models. Typically, learned models on selective copying learn to output the correct token at each position in the context, while the construction only provides the correct token in the last position. In contrast, the associative recall with decoding construction outputs the correct token at each position in the context, similar to learned models. This difference can be understood in when a task uses fixed positional differences. The selective copying construction uses a fixed mechanism to look-up a distance away from the last token, while learned models learn a more general relative positioning. Associative recall with decoding does not use relative positions, leading to a construction that more readily works at every token position.

# E    Experiment Details

## E.1    Expressivity Experiments

All experiments had identical learning rate sweeps. Additionally, all experiments were trained to convergence using an AdamW optimizer. We used 100 steps of warm-up followed by a linearly decaying learning rate. The learning rate sweep was over the maximum learning rate.

Experiments were ran 11 times, with the mean performance shown along with the 10 and 90 percentiles as error bars.

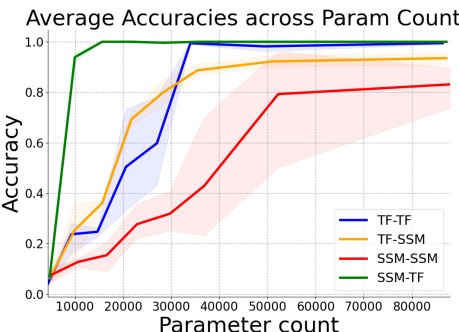

*Figure 11.* Results of training the same architecture as the other, smaller vocabulary experiments, except with more tokens. The left figure shows results for a vocabulary of size 200, and the right figure shows results for a vocabulary of size 1000.

The positional encodings used where RoPE (although similar behavior was observed for learned positional encodings). Transformers are made from GPTNeoX (Black et al., 2022), and SSM layers are coming from Mamba.

Experiment sweeps were conducted across token dimension. The Mamba layers contained more parameters than the Transformer layers, hence why some figures stop sooner for pure Transformers than for hybrids, and pure SSMs extend beyond the hybrids.

Experiments at this scale were done with input length 100, unless otherwise specified. Investigating the behavior of these different architectures further, some other parameters were varied. Specifically, we were interested in how the number of heads, or the side of the Mamba state, affects the performance of these models. A sweep of learning rates were tested from 1e-4 to 1e+0 by factors of $\sqrt{10}$. All experiments were trained to convergence, typically with over 4x the compute after the loss plateaus.

These models were also trained as seq-to-seq tasks rather than autoregressively. This was to prevent certain simple tasks from having simpler properties, such as becoming cyclic, which the models could learn instead.

**Selective Copy.** For this task, we used number tokens 5 through 10 and 26 vocabulary tokens. Larger vocabulary sizes are explored below, although the behavior is similar. The default token dimension is 12; this was the scale at which separations could be seen. At the scale of modern LLMs, this task is easily learned.

**Associative Recall with Decoding.** This task proved less learnable than the others. The target dimensions swept were between 24 and 768. We used a bit sequence length of 5, so a vocabulary size of $2^5 = 32$ beyond these two bits. These experiments also specifically used three layer models rather than two layer ones; all two layer models never learned anything.

**Multi-Key Associative Recall.** For this task, we used a query length of 2 and a vocabulary size of 8. This very small vocabulary came from needing to have seen the target pair in the context (of length 100). Any larger than $8^2 = 64$ keys would make failure to see the key occur with high probability. Similarly to selective copying, token dimensions were kept small, around 12.

**Needle in a Haystack.** We used a vocabulary of 100 tokens, plus two marker tokens to indicate the position of the needle, and where the needle should be outputted.

### E.2 Additional Experiments

**Selective Copy.** We also tried larger vocabulary sizes for this task; this is something not expected to substantially change behavior. While a larger vocabulary is definitely possible, results were not significantly different for vocabularies of 200 or 1000. See Figure 11.

Results for changing number of heads and state dimension can be see in Figure 12. As expected, we observe that increasing the number of heads improves performance. However, these same results may not be observed if simply increasing the number of heads in a standard implementation of a transformer block. This is because typically heads have a lower embedding dimension than the residuals. This keeps the number of parameters the same, but decreases what each individual head can do on its own. When conducting this experiment in that setting, all performances above one head dropped to

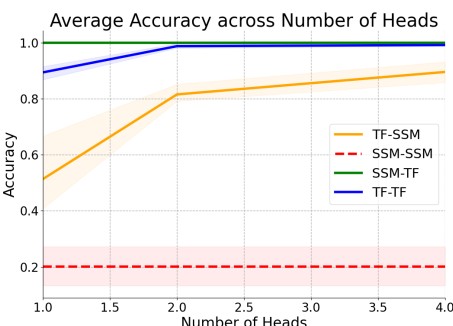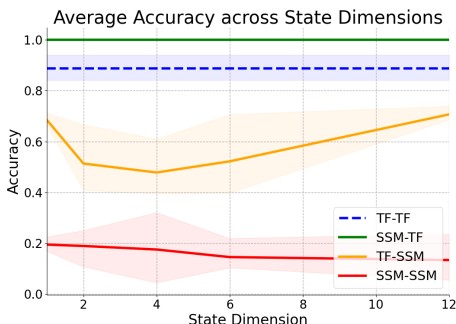

*Figure 12.* Results from training small models on Selective Copy. (a) changes the number of heads, and (b) increases the state dimension as described in Mamba. Defaults are token dimension 12, number of heads 1, and state dimension 1. Error bars are 0.1 and 0.9 quantiles around the mean.

random. Here, we instead kept every head the same dimension as the original model and increased the number of parameters with increasing the number of heads.

The same does not hold for increasing the dimension of the state inside of the Mamba model. In this case, there was a general degradation in performance with greater state. We believe this is mostly due to optimization issues for Mamba.

Additionally, we trained these small models on a more adversarial distribution, where according to the developed theorems, both the SSM and Transformer should perform poorly on half of the instances. For this distribution, half of the data points maintained a number token as their final token. This made the task difficult for SSMs to perform since they would need sufficient state to store all possible outputs for a different final token each time. On the other hand, half of the instances had very sparse number tokens, meaning that a Transformer with a small window for its attention would likely have the most recent number token outside of its context, hindering its performance. However, we should still expect the hybrid to perform well as both of these issues are mitigated.

As we can see in Figure 13, the hybrid still does outperform both of the pure models, and the reverse hybrid. However, unlike the uniform distribution seen in Figure 4, the hybrid performs much better in this environment. This could be because of many reasons, one being that in a uniform distribution, it might be harder for the SSM to decern the task, whereas when half of the instances have the number token as the final token, the pattern may make itself more apparent. The Transformer still performs around as well in this distribution, possibly indicating that in both cases number tokens frequently are outside of the windows for the Transformer.

When expanding the available context window to these models, we observe two important things. First, Transformers tend to get *worse*. This is different from what is predicted by our theory, indicating that there is an additional learnability difficulty for Transformers with larger contexts which hybrids avoid. Second, with large window sizes, the hybrid also frequently suffers, learning no better than the Transformer. This, again, is due to learnability issues rather than expressivity ones.

**Multi-Key Associative Recall.**

Increasing the number of heads and the size of the state tends to show both what is expected from the theory as well as the optimization issues more prevalent in the Selective Copy experiments. First, performance drops because of optimization issues, but after sufficient heads/state is added, performance again begins to grow.

We can also see that when the side of the state increases, the pure SSM model grows in performance from a state dimension of 2 to a state dimension of 6, before decreasing again, likely again from these optimization issues.

Also similar to selective copying, we see that as window size increases, the performance of the Transformer degrades, rather than improving. This is still due to learnability issues for long contexts with these Transformers, which hybrids mitigate.

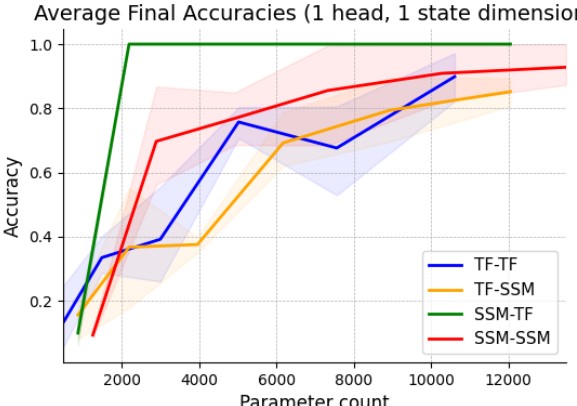

*Figure 13.* Selective Copy trained on an adversarial distribution, where half of the instances have a number token as their final token (hard for SSM), and half of the instances embed their last number token early in the sequence (hard for Transformers). This distribution is empirically easier for SSMs than uniform.

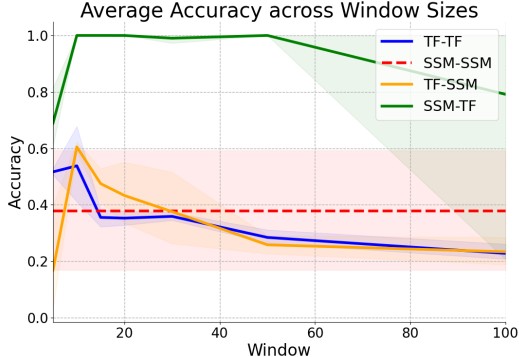

*Figure 14.* Results from training small models on Selective Copy. This changes the window of the context available to the model.

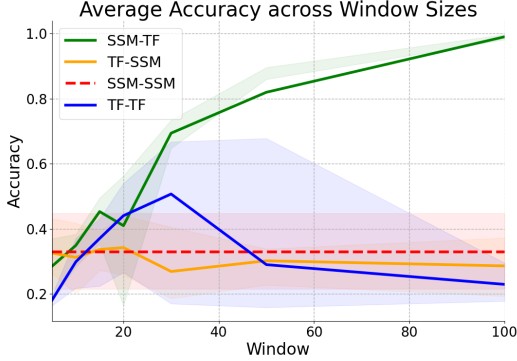

*Figure 15.* Results from training small models on Multi-Key Associative Recall. This changes the window of the context available to the model.

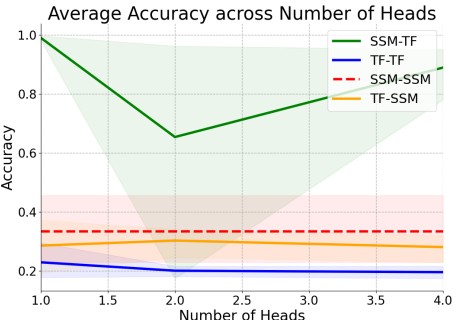 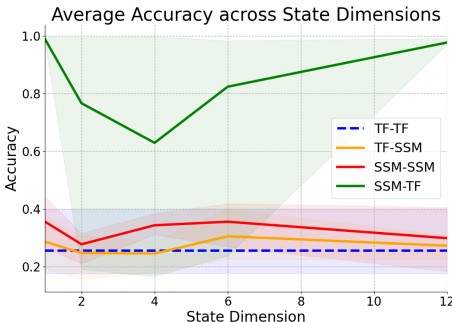

*Figure 16.* Results from training small models on Multi-Key Associative Recall. (a) changes the number of heads, and (b) increases the state dimension as described in Mamba. Defaults are token dimension 12, number of heads 1, and state dimension 1. Error bars are 0.1 and 0.9 quantiles around the mean.

