# OpenReview forum: "Expressivity-Efficiency Tradeoffs for Hybrid Sequence Models"
_ICML.cc/2026/Conference — ICML 2026 spotlight_

### Official Review · Reviewer_xNV5 · 2026-02-26

**Soundness:** 4
**Presentation:** 3
**Significance:** 4
**Originality:** 4
**Overall Recommendation:** 5
**Confidence:** 4

**Summary:**

The paper studies the expressivity and parameter-efficiency of hybrid models (mixing transformer and state-space model (SSM) layers) compared to pure models on a suite of synthetic tasks. The main results reveal that hybrid models can be substantially more parameter-efficient at solving concrete tasks than any pure (transformer or SSM) model. Concretely, the paper studies function composition and its use for solving long-context tasks. Pure architectures struggle at solving these tasks (requiring large parameter counts or state sizes), while hybrid models can leverage the strengths of the individual layers to solve them efficiently.
The authors complement the theoretical findings with empirical analysis on these synthetic tasks, showing that hybrid models indeed learn the studied tasks better and exhibit better length generalization.

**Compliance With Llm Reviewing Policy:**

Affirmed.

**Final Justification:**

The paper makes a valuable and well-executed contribution to the study of hybrid models, and the rebuttal addressed my concerns.

**Key Questions For Authors:**

1. This is not central to the understanding of the results, but I just wanted to confirm my understanding: If I understand correctly, Lemma 3.5 requires the individual layers of the SSM to be relatively expressive (being able to simulate any fixed number of layers at once). If we constrain a single layer to, e.g., implement a single-hidden layer MLP, this Lemma does not apply anymore, right?
2. The results (particularly the limitations of pure SSMs) rely on the finiteness of the state space. Do you have any intuitions on how these results would generalize to growing precision (e.g., log-precision) models?
3. Your constructions focus on using SSM layers before transformers, and the empirical results suggest this is the right way to hybridize models. Besides the fact that your constructions go through with this, do you have any intuition behind what is special about this way of stacking layers?

**Limitations:**

yes

**Strengths And Weaknesses:**

- Significance: The paper tackles a very lively and active research area of sub-quadratic architectures and how to combine them with the strengths of transformers. I think this is a very exciting area to study, and the paper contributes valuable insights into the architecture design.
- Significance, Originality: This is one of the first results theoretically studying the strengths and affordances of model *hybridization* compared to pure architectures, showing concrete tasks where hybrid models can outperform pure architectures. This is both intuitively understandable and aligned with empirical findings of large-scale studies.
- Presentation: The presentation is very clear and easy to follow. The tasks are described and motivated well, and the paper flows well from theoretical results (limitations of pure models to strengths of hybrid models) to empirical results that augment the theoretical ones.
- Presentation, Significance: The considered tasks are both specific enough to distinguish hybrid and pure models (the first such tasks to my knowledge) and relevant to real-world applications and connect well to existing benchmarks.
- Soundness, Presentation: The theoretical results are well-explained (with a very appropriate mix of intuition in the main text and proofs in the appendix), well-motivated, relevant, and seem correct.
- Significance: The empirical results and the experimental design seem thorough and provide additional value. For example, they confirm the theoretical results and expand on them in terms of learnability, and they also test tasks outside of the scope of the theoretical results, which is welcome. Moreover, they confirm the interesting phenomenon that the order of hybridization matters—putting the SSM layer before the transformer seems to matter.
- Presentation: Three small nits:
	1. While it is useful and reassuring that the theoretical constructions were implemented, I don’t think including them in the empirical results section is strictly necessary, as their better performance can be inferred from the theoretical results immediately.
	2. While I’m not entirely familiar with the existing literature on this, I don’t know if the name “function composition” is the most suitable for the overarching task (Def. 3.1). At most, we have one layer of composition (composing $u$ and $v$ with $F$), while, to me, function composition would imply studying arbitrary-deep compositions of functions.
	3. Similarly, I believe your definition of local sensitivity is exactly the opposite of the standard one in formal language theory (see, e.g., Jäger, Gerhard, and James Rogers. "Formal Language Theory: Refining the Chomsky Hierarchy."). Maybe using a different term would resolve the clash?

---

> ### Author Rebuttal · Authors · 2026-03-31
>
> We thank the reviewer for noting the novelty and significance of our work. We provide our responses to all of the reviewer's comments next:
>
> - ***On the implemented constructions.*** (W1) We aim to include this for two reasons: **for the assurance of the constructions working as desired** (as mentioned) and **to show the dimension needed to implement the task, with this vocabulary size and sequence length is small**. The latter arises as an issue since the $O(\cdot)$ and $\Omega(\cdot)$ can hide large constants, which we want to show are small.
>
> - ***Nomenclature choices.*** (W2, W3) We have changed function-composition to **“structured two-layer function composition (S2FC)”**, and R-local sensitivity to **“R-global sensitivity”** (mirroring the definition in formal language theory) to make the names more appropriate.
>
> - ***Lemma 3.5 expressivity requirements.*** (Q1) To clarify, we agree that Lemma 3.5 holds for incredibly expressive SSM layers. For instance, if multiple Mamba layers were stacked in practice, we should not immediately expect these to be equivalent to a single Mamba layer with greater state as a result of the parameterization. Importantly, the lemma establishes that **any SSM, no matter how expressive, cannot achieve beyond what its state allows**.
>
> - ***Changing the precision of the SSM state.*** (Q2) Our results involve a value proportional to the number of bits in the state, so **doubling precision is similar to doubling the dimension, as both double the number of available bits**. This holds for an SSM that is expressive enough to treat every bit separately; real SSMs likely won’t see such a great improvement since they do not treat every bit separately.
>
> - ***On layer ordering.*** (Q3) There is an intuition for the hybrid order for function-composition tasks. Their computation first requires $v$ followed by computing $F$ with the local context $u$. This is the wrong order for the TF->SSM hybrid, where an SSM is better with $v$ and the transformer is better with $u$. While it is an interesting problem, deriving results that precisely address layer-ordering for models with a large number of layers is beyond the scope of this work. However, we anticipate tackling this in future work.

---

> > ### Author Rebuttal · Reviewer_xNV5 · 2026-03-31
> >
> > The rebuttal addressed my questions. Thank you for the helpful explanations and for modifying the terminology. I maintain my positive evaluation and hope the paper gets accepted.

---

> > > ### Author Response · Authors · 2026-04-07
> > >
> > > Thank you for your careful reading of our rebuttal, and we are glad that our responses resolved your concerns, especially with regard to terminology. We sincerely appreciate your support for our work!

---

### Official Review · Reviewer_tUDA · 2026-03-12

**Soundness:** 4
**Presentation:** 4
**Significance:** 4
**Originality:** 3
**Overall Recommendation:** 5
**Confidence:** 5

**Summary:**

This work studies the benefits of hybrid models (SSM + Attention based) over pure SSM or pure attention based models in terms of efficiency and expressivity. They present theoretical results validating that hybrid models can achieve high efficiency with good expressivity on synthetic tasks while using less resources than standard models. They empirically corroborated they results over a more extensive set of tasks, high context length and out of distribution samples. They show that, in general, hybrid models outperform standard transformers or SSMs with the same amount of parameters.

**Compliance With Llm Reviewing Policy:**

Affirmed.

**Final Justification:**

Authors answered my questions and my score is already positive. I think this is a solid work and match ICML expectations.

**Key Questions For Authors:**

1) Is there a reason that transformer layers should come later than SSM layers? Intuitively I understand that SSM’s layers can retrieve relevant information from longer context but maybe this can be formalized so it is never a good idea to add transformer layers before SSM’s ones.
2) Is there a way to extrapolate the simple tasks you presented into more complex reasoning tasks?
3) Is there a formal reason of why hybrid models have a better generalization capacity than standard models?
4) Can you give details on how do you test longer sequences and which tasks did you choose for this analysis? Also, what is the 12 layer architecture you used for the hybrid model in OOD analysis?

**Limitations:**

yes

**Strengths And Weaknesses:**

Strengths
- This paper is exceptionally well written, has strong theoretical results that shows classes of compositional problems that can not be efficiently solved by SSMs and Transformers but can actually be efficiently solved by hybrid models.
- All theoretical results are backed by appropriate empirical results, with appropriate notation and clear visualizations
- Authors further analysed the generalization capacity of hybrid models against standard models, showing that even in this
- This work is a considerable step towards a proper expresivity and efficiency separation between hybrid models and standard SMM’s or transformers

Weaknesses
- Experiments and theoretical results were developed over a fixed set of really simple synthetic tasks
- Even though generalization analyses are interesting, very few details are given on the models they used and how they overcame vocabulary restrictions when extending context
- No code for replication is given

minor comments:

L57 - of the problem to solve the problem - Sounds awkward
L325 - Typo: “accuracy if measured”- should be “is measured"

---

> ### Author Rebuttal · Authors · 2026-03-31
>
> Thank you for your careful read and the excellent suggestions. Each of the comments is addressed below.
>
> - ***Code for reproducibility.*** (W3) In the submitted paper, **there is a link to an anonymized repo** on the first page of our submission below the abstract, L53-L54. This contains all of the code for our work.
>
> - ***The simplicity of the chosen tasks.*** (W1) While the tasks are simple, **they measure capabilities that are key components for fundamental language modeling**. Mamba [1] and other works have established that a selection mechanism (and specifically selective copying), where relevant tokens need to be filtered from the context, is critical to language modeling. **Both selective copying and associative recall with decoding require the selection mechanism**, filtering numbers and bits respectively. Our choice of these tasks is both inspired by prior work on SSMs and architecture design and by their importance and popularity in mechanistic interpretability.
>
>
> - ***On layer ordering.*** (Q1) Our central goal for this work is to show that hybrid models outperform their pure counterparts. While it is an interesting problem, formalizing the details of precise layer-ordering for models with a large number of layers does not fit our initial objectives for this work. On top of the intuition suggested, there is another intuition for the hybrid layer order for two-layer models that may extend to larger number of layers, which we will tackle in future work. Specifically, *function-composition computation is inherently serial*, first requiring $v$ followed by computing $F$ using $u$. It becomes hard for a TF->SSM model to perform this task, since at least one layer is required to perform the part of the task that is hard for that layer type.
>
> - ***Extrapolating to reasoning tasks.*** (Q2) This is an excellent point. Indeed, note that the tasks we have selected are required components for multi-hop reasoning. Many such reasoning tasks involve facts that must be remembered along with a complex operation based on these facts. **These readily factor into $v$ for facts, $u$ for local context, and $F$ for the complex operation**.
>
> - ***Formal reason for generalization.*** (Q3) The reviewer asks if there is a formal reason to expect hybrid models length-generalize better than non-hybrids. The experiments we carried out for length generalization are meant to show that **our select tasks probe more than simply the separation in expressivity between these model architectures**. We do not attempt to formally prove this in this work, but we believe that formally studying this type of generalization in the context of hybrids is an excellent direction for our future work.
>
> - ***Length experiment details.*** (Q4) The details are in Section 5.3 lines 378-432, including the chosen task; see as well the provided code in `mini/models/hybrid.py` for the structure of the hybrid as well as `mini/data_utils.py` lines 369-411 for context packing and lines 230-248 for data generation. We also recap these here for convenience:
> -- _Architecture_: The used architecture is a hybrid where every other layer is different, starting with an SSM layer. This model had a total context length of 200.
> -- _Task_: We used associative recall with decoding, since it proved the most difficult for smaller models. We set the number of bits to be 4.
> -- _Train Sequences_: Samples of length 200 were constructed by concatenating examples of length 20 to 50, selected at random, concatenated together with <bos> and <eos> tokens. The proportion of bits in these sub-sequences was set to 0.2, so most sequences should include a number of bits within them.
> -- _Eval Sequences_: Each sequence had length $L$, still with a 0.2 proportion of bits, where $L$ was varied from 20 to 200.
> -- _Evaluation_: Accuracy is measured as the average accuracy on all tokens that have a valid output. This omits all tokens that do not have sufficient tokens before them in the sequence.
> We have added additional details to the end of Appendix E in our revised manuscript.
>
> [1] Gu, A. and Dao, T. Mamba: Linear-time sequence modeling
> with selective state spaces. In First Conference on Language Modeling, 2024.

---

> > ### Author Rebuttal · Reviewer_tUDA · 2026-03-31
> >
> > Thank you for answering my questions. I think you have a nice paper so I will keep my score.

---

> > > ### Author Response · Authors · 2026-04-07
> > >
> > > Thank you for carefully reviewing our responses. We appreciate your positive view of our work.

---

### Official Review · Reviewer_VbY5 · 2026-03-13

**Soundness:** 4
**Presentation:** 4
**Significance:** 4
**Originality:** 3
**Overall Recommendation:** 5
**Confidence:** 4

**Summary:**

This paper studies the expressivity-efficiency tradeoffs inherent to hybrid sequence models that combine Transformer (attention) layers with state-space model (SSM) layers. The authors define a family of "function-composition" tasks (Definition 3.1) and prove lower bounds showing that pure SSMs require large state space (Theorem 3.3) and pure sliding-window Transformers require large window size (Theorem 3.7) to solve such tasks. They then construct specific hybrid models for two tasks, selective copying (Theorem 4.3) and associative recall with decoding (Theorem 4.6) that achieve sublinear memory and polylogarithmic parameter counts. Empirical experiments on these and two additional tasks (MKAR, needle-in-a-haystack) validate the theoretical claims and demonstrate that learned hybrids outperform pure models with up to 6x fewer parameters, with additional benefits in length generalization and OOD robustness.

**Compliance With Llm Reviewing Policy:**

Affirmed.

**Final Justification:**

I thank the authors for their responses. The rebuttal addressed the concerns I had and I hope the paper gets accepted.

**Key Questions For Authors:**

1. The key theoretical claim is that Transformers need large windows but full-attention Transformers trivially have window = L, satisfying the bound. Can you provide any theoretical result that applies to full-attention Transformers, or do you view the hybrid advantage as fundamentally about inference efficiency (sublinear KV cache) rather than expressivity?
2. Authors observe that Transformers get worse with larger windows in several experiments (Fig 14–15), which contradicts the expressivity theory. This is attributed to "learnability issues." Can you say more about what these optimization failures look like? Is it attention dilution, loss landscape issues, or something else?
3. Is there a theoretical reason why SSM -->TF dominates TF --> SSM in your framework, beyond the construction assuming this ordering?

**Limitations:**

1. The theoretical framework is inherently restricted to sliding-window Transformers. Full-attention models are untouched by Theorem 3.7, so the hybrid advantage the paper formalizes is really about inference efficiency under a fixed window budget, not expressivity in the usual sense. The paper does not formalize this efficiency-oriented reading.
2. The function-composition framework has uneven coverage over the paper's own task suite. It does not predict a hybrid-over-Transformer separation for MKAR (v is empty), yet one is observed empirically. For needle-in-a-haystack, it predicts no SSM hardness, and none is observed. The framework therefore explains some but not all of the reported results, and the paper offers no account of what additional structure would close the gap.
3. The expressivity results say nothing about learnability, and the experiments repeatedly surface this disconnect (Transformers degrading with larger windows, non-monotonic gains from increasing SSM state) without resolving it. Since the paper's practical message depends on learned hybrids, not constructed ones, this is a scope limitation that directly affects how much the theory can explain about the experiments.

**Strengths And Weaknesses:**

Strengths
1. The function-composition abstraction (Definition 3.1) is a clean way to capture tasks where one component extracts context (u) and another controls the output (v). The two assumptions (3.2 for SSM hardness, 3.6 for Transformer hardness) are intuitive and well-articulated: one captures the information-bottleneck of finite state, the other captures the locality limitation of bounded windows. The framework is general enough to subsume multiple existing synthetic tasks.
2. Authors provide weight-level constructions for both selective copying (2-layer Mamba+Attention) and associative recall with decoding (3-layer Mamba+2xAttention).
3. The three-claim structure (C1 construction verification, C2 learnability, C3 generalization) provides a clear logical progression. The inclusion of MKAR and needle-in-a-haystack as additional benchmarks beyond the theoretically analyzed tasks shows some effort toward generality.
4. The SSM lower bound via Fano's inequality and mutual information (Theorem 3.3 proof) is technically sound. The Transformer lower bound via the locality argument (Theorem 3.7) is simple but appropriate for sliding-window models.

Weaknesses
1. The Transformer model is severely restricted, making the lower bound weak. Assumption 3.6 and Theorem 3.7 apply only to sliding-window Transformers. The proof is essentially trivial, if the output depends on tokens outside the cumulative window, the model fails. This is a limitation of windowed attention, not of Transformers in general. Full-attention Transformers (which are the dominant practical architecture) are not covered by this lower bound. The paper does test full-attention Transformers empirically (Fig 7 for NH), and in that case the Transformer does fine. The theoretical contribution regarding Transformer limitations is therefore narrow.
2. The gap between theory and practice is large and under-discussed. The theoretical constructions assume exact weight settings, binary embeddings, and specific structural choices. The experiments use standard training (AdamW, learning rate sweeps) on models with very small embedding dimensions (d=12 for selective copying, d=24–768 for AR with decoding). The paper acknowledges optimization difficulties multiple times. Authors claim that transformers get worse with larger windows (contradicting the theory), and their performance degrades from increasing state dimensions, etc. However the authors treat these as secondary observations rather than engaging with why the theoretical predictions fail to hold directionally in some regimes.
3. There is no comparison with other hybrid configurations (interleaved SSM-TF-SSM-TF, or more than 2–3 layers). The "reverse hybrid" (TF→SSM) is tested, which is good, but multi-layer hybrids with different SSM/TF ratios are not explored.

---

> ### Author Rebuttal · Authors · 2026-03-31
>
> We thank the reviewer for their careful read and for noting the clarity in our formulation. We respond to each comment below.
>
> - ***On full-attention Transformers.*** (W1, Q1) The reviewer asks if and how full-window Transformers fit into our theory. **In our analysis, full-window Transformers are a special case of windowed Transformers**. Specifically, full-window Transformers consume the most memory of all such architectures; comparing hybrids against these models alone shows even more dramatically that a hybrid can perform better with significantly less memory. The reason for our analysis being at this level of generality (covering any sliding window length) is to show that hybrid models of SSMs and Transformers, including with a sliding window, _need less memory than pure models of either architecture for the same performance_. Indeed, this is particularly relevant at the frontier scale where memory constraints are of great interest to larger model design.
>
> - ***Empirical-Theoretical differences.*** (W2, Q2) **None of the experimental results contradict our theoretical findings**: the bounds we derive represent upper limits on accuracy; performance below these limits agrees with the theory. The purpose of the experimental results is to (1) gauge how hybrids and non-hybrids perform when training is added as an ingredient, and (2) to derive additional insights. The reviewer's observation fits within (2):
> -- _Window Size_: The issue with larger windows appears to be attention dilution: the Transformers have difficulty identifying the most recent number token rather than the few most recent.
> -- _SSM Dimension_: For the SSM state size, we did not observe improved performance. We suspect that the particular choice of SSM may be related to this.
> -- _Number of Heads_: We modified the Transformer implementation to create square heads rather than splitting a single square head, as is common in practice. The resulting performance of Transformer models now monotonically increases. Here are the precise results (measured as average performance):
>
> |Model|1 Head|2 Heads|4 Heads|
> |-|-|-|-|
> |SSM -> SSM|0.20|0.20|0.20|
> |SSM -> TF|1.0|1.0|1.0|
> |TF -> SSM|0.52|0.82|0.89|
> |TF -> TF|0.88|0.96|0.98|
>
> - ***On layer ordering.*** (W3, Q3) Our central goal for this work is to show that **hybrid models outperform their pure counterparts**. If any hybrid provably out-performs non-hybrids, we have achieved our primary goal. While it is an interesting problem, pinning down the details of precise layer-ordering for models with a large number of layers does not fit our initial objectives for this work. We do note that there is an intuition for the hybrid layer order for two-layer models that may extend to larger number of layers, which we will tackle in future work. Specifically, function-composition computation is inherently serial, first requiring $v$ followed by computing $F$. For a TF->SSM hybrid, either the TF has to compute $v$ (which we establish requires large memory) or the SSM does. This SSM would then also need to compute $F$, which requires $u$ (which requires large memory as well).

---

> > ### Author Rebuttal · Reviewer_VbY5 · 2026-04-05
> >
> > I thank the authors for the reponse. I will keep my score and I hope the paper gets accepted.

---

> > > ### Author Response · Authors · 2026-04-07
> > >
> > > Thank you for carefully reviewing our responses. We greatly appreciate your support for our work!

---

### Decision · Program_Chairs · 2026-04-30

**Decision:**

Accept (spotlight)

**Comment:**

All reviewers enthusiastically support acceptance and all raised issues have been successfully addressed during rebuttal. I believe the paper makes important contributions to the growing field of hybrid architectures, both theoretically and empirically. I strongly recommend acceptance, potentially as an oral presentation.